# NEIL1 and NEIL2 DNA glycosylases modulate anxiety and learning in a cooperative manner in mice

Gunn A. Hildrestrand [1,15], Veslemøy Rolseth [1,11,15], Nicolas Kunath [2,15], Rajikala Suganthan[1], Vidar Jensen[3], Anna M. Bugaj[2], Marion S. Fernandez-Berrocal [2], Sunniva B. Sikko [2], Susanne Vetlesen[1], Anna Kuśnierczyk [4], Ann-Karin Olsen [5,6], Kristine B. Gützkow[5,6], Alexander D. Rowe[1,12], Wei Wang[2], Olve Moldestad[7,13], Monica D. Syrstad[1], Geir Slupphaug[4], Lars Eide [8], Arne Klungland [1,14], Pål Sætrom [2], Luisa Luna[1], Jing Ye [2], Katja Scheffler[2,9,10,16] & Magnar Bjørås [1,2,16 ✉]

Oxidative DNA damage in the brain has been implicated in neurodegeneration and cognitive decline. DNA glycosylases initiate base excision repair (BER), the main pathway for oxidative DNA base lesion repair. NEIL1 and NEIL3 DNA glycosylases affect cognition in mice, while the role of NEIL2 remains unclear. Here, we investigate the impact of NEIL2 and its potential overlap with NEIL1 on behavior in knockout mouse models. $Neil1^{-/-}Neil2^{-/-}$ mice display hyperactivity, reduced anxiety and improved learning. Hippocampal oxidative DNA base lesion levels are comparable between genotypes and no mutator phenotype is found. Thus, impaired canonical repair is not likely to explain the altered behavior. Electrophysiology suggests reduced axonal activation in the hippocampal CA1 region in $Neil1^{-/-}Neil2^{-/-}$ mice and lack of NEIL1 and NEIL2 causes dysregulation of genes in CA1 relevant for synaptic function. We postulate a cooperative function of NEIL1 and NEIL2 in genome regulation, beyond canonical BER, modulating behavior in mice.

[1] Department of Microbiology, Oslo University Hospital and University of Oslo, N-0424 Oslo, Norway. [2] Department of Clinical and Molecular Medicine, Norwegian University of Science and Technology, N-7491 Trondheim, Norway. [3] GliaLab, Department of Molecular Medicine, Institute of Basic Medical Sciences, University of Oslo, N-0317 Oslo, Norway. [4] Proteomics and Modomics Experimental Core Facility (PROMEC), Norwegian University of Science and Technology, N-7491 Trondheim, Norway. [5] Department of Molecular Biology, National Institute of Public Health, N-0456 Oslo, Norway. [6] Centre of Excellence "Centre for Environmental Radiation" (CERAD), NMBU, N-1433 Ås, Norway. [7] Laboratory of Cellular Neurophysiology and Ion Channel Function, Department of Molecular Medicine, Institute of Basic Medical Sciences, University of Oslo, N-0317 Oslo, Norway. [8] Department of Medical Biochemistry, Oslo University Hospital and University of Oslo, N-0424 Oslo, Norway. [9] Department of Neuromedicine and Movement Science, Norwegian University of Science and Technology, N-7491 Trondheim, Norway. [10] Department of Neurology, University Hospital of Trondheim, N-7006 Trondheim, Norway. [11]Present address: Department of Forensic Sciences, Division of Laboratory Medicine, Oslo University Hospital, N-0424 Oslo, Norway. [12]Present address: Department of Newborn Screening, Division of Paediatric and Adolescent Medicine, Oslo University Hospital, N-0424 Oslo, Norway. [13]Present address: Department for Rare Disorders, Division of Paediatric and Adolescent Medicine, Oslo University Hospital, N-0424 Oslo, Norway. [14]Present address: Department of Biosciences, University of Oslo, N-0371 Oslo, Norway. [15]These authors contributed equally: Gunn A. Hildrestrand, Veslemøy Rolseth, Nicolas Kunath. [16]These authors jointly supervised this work: Katja Scheffler, Magnar Bjørås. ✉email: magnar.bjoras@ntnu.no

Cells in tissues and organs are continuously subjected to oxidative stress originating both from exogenous and endogenous sources such as reactive oxygen species (ROS), ionizing radiation, UV radiation, and chemicals, amongst others[1]. The brain is especially susceptible to oxidative stress due to a high metabolic rate, low levels of antioxidant enzymes, and high levels of iron[2–4]. Thus, repair of oxidative damage in the genome of postmitotic neurons is supposed to be critical for proper brain function[5–7]. The hippocampus is a brain area critical for learning and memory formation and is also involved in anxiety[8–12]. Increasing evidence shows that oxidative stress and defective DNA repair affects the hippocampus and leads to cognitive impairment[13–16]. Oxidative stress has also been implicated in depression and anxiety[14,16–18]. In mammalian cells, oxidative DNA damage is predominantly repaired via the base excision repair (BER) pathway (reviewed in[19]) and enzymes in this pathway have been shown to be important for protection against neuronal cell death following induced ischemic brain damage[20–23]. BER is initiated by DNA glycosylases, which recognize and remove small base lesions (reviewed in[24,25]). To date, eleven mammalian DNA glycosylases have been identified. NEIL1 and NEIL2 are two of five DNA glycosylases specific for oxidative base lesions and the substrate specificities for these DNA glycosylases are partially overlapping. NEIL1 has broad substrate specificity and removes both pyrimidine- and purine-derived lesions such as 4,6-diamino-5-formamidopyrimidine (FapyA), 2,6-diamino-4-hydroxy-5-formamidopyrimidine (FapyG), guanidionhydantoin (Gh), spiroiminodihydantoin (Sp), and thymine glycol (Tg) from DNA. NEIL2 primarily removes oxidation products of cytosine such as 5-hydroxy-cytosine (5-ohC) and 5-hydroxy-uracil (5-ohU), also excised by NEIL1[19,26–28]. *NEIL1* and *NEIL2* mRNA is homogeneously distributed and ubiquitously expressed in human and murine brain, indicating a role of NEIL1 and NEIL2 in DNA maintenance in most areas of the brain[29]. Previous studies of mice lacking NEIL1 revealed a metabolic phenotype with variable penetrance, impaired memory retention and defects in olfactory function, as well as increased sensitivity to ischemic brain injury[23,30–32]. No overt phenotype has been reported for NEIL2-deficient mice, but they were shown to accumulate oxidative damage in transcribed regions of the genome with age[33]. Recently, we reported no accumulation of DNA damage or mutations and no predisposition to cancer development in mice lacking both NEIL1 and NEIL2[34].

In the present study, we used mice deficient in NEIL1 and/or NEIL2 DNA glycosylases to elucidate the roles of these enzymes in behavior and cognition (i.e., activity, anxiety, learning, and memory) and to study their impact on genome stability, gene expression and electrophysiological features in the hippocampus. Our study revealed an altered behavioral phenotype in NEIL1/NEIL2-deficient mice, which was accompanied by differential regulation of genes relevant for synaptic function and instability of NMDA-receptor architecture in the hippocampus. No accumulation of DNA damage or mutations point to a NEIL1/NEIL2-dependent regulation of synaptic factors that is not explained by the enzymes' function in DNA repair but rather a noncanonical contribution to gene regulation.

## Results

### Hyperactivity, reduced anxiety, and improved learning in $Neil1^{-/-}Neil2^{-/-}$ mice. The functional consequences of inactivating the *Neil1* and *Neil2* genes were investigated by behavioral studies in adult (6-month-old) male mice (Fig. 1a). General activity levels were examined in an open field maze (OFM) and anxiety was monitored by using an elevated zero maze (EZM). Hippocampal functions such as spatial learning and memory

were investigated by using the Morris water maze (MWM). The $Neil1^{-/-}Neil2^{-/-}$ mice displayed hyperactive behavior both in the OFM and the EZM by being more mobile (Fig. 1b, f) and covering an increased distance when exploring the two mazes (Fig. 1c, g), compared to single knockout (KO) and wild-type (WT) mice. In both mazes, they entered the center and open area zones more frequently than the other mice (Fig. 1d, h). In the EZM they also spent more time in the open area zones, which is indicative of reduced anxiety (Fig. 1i). All these characteristics were also observed in the single KO mice, however, to a lesser extent. In the MWM, mice of all genotypes swam well and learned the position of the hidden escape platform, as indicated by reduced latencies to escape the water during training, days 1 to 4 (Fig. 1j). Surprisingly, the $Neil1^{-/-}Neil2^{-/-}$ mice needed less time to localize the escape platform during training days 2–4, compared to WT mice, suggesting enhanced learning capacity in the double KO (DKO) mice. The $Neil2^{-/-}$ mice also showed a tendency to improved learning; nonetheless, this was not statistically significant (Fig. 1j). During the first retention trial (day 5), all genotypes showed similar occupancies at the target quadrant (Fig. 1k), suggesting no substantial differences in spatial memory in any of the genotypes. During the second retention trial (day 12), the $Neil1^{-/-}Neil2^{-/-}$ mice appeared less decisive, shown as decreased preference for the target zone (Fig. 1k) and a tendency to search further away from the platform (Fig. 1l). However, these differences were not statistically significant. Thus, the probe tests suggest that memory (as measured in the MWM) was not impaired in any of the mutants. In line with a previous report[30], the $Neil1^{-/-}$ mice weighed significantly more than the WT mice (Supplementary Fig. 1). The $Neil1^{-/-}$ mice were also significantly heavier than the $Neil2^{-/-}$ and $Neil1^{-/-}Neil2^{-/-}$ mice (Supplementary Fig. 1). The increased weight did not seem to affect the activity level of the $Neil1^{-/-}$ mice (Fig. 1b, c, f, g). Overall, the behavioral tests revealed an altered behavioral phenotype in the $Neil1^{-/-}Neil2^{-/-}$ mice shown as hyperactivity, reduced anxiety-like behavior, and improved learning.

### No change in steady-state levels of oxidative DNA base lesions and no hypermutator phenotype in $Neil1^{-/-}Neil2^{-/-}$ hippocampus. As the hippocampus is one of the critical brain areas involved in anxiety as well as learning and memory, we assessed the effect of NEIL1 and/or NEIL2 deficiencies on hippocampal DNA integrity. We applied three different methods on adult (4–6-month-old) male mice (Fig. 2a). First, the bulk level of the oxidative DNA base lesion 5-ohC, a substrate for both NEIL1 and NEIL2, in hippocampal genomic DNA was measured by mass spectrometry (HPLC-MS/MS). The results showed no significant differences in global 5-ohC levels between the four genotypes (Fig. 2b). Second, the alkaline comet assay was used to analyze DNA damage, including strand breaks, at the single-cell level. The mutants had not accumulated more DNA damage than WT mice; on the contrary, the hippocampi of $Neil2^{-/-}$ and $Neil1^{-/-}Neil2^{-/-}$ mice displayed reduced levels of strand breaks genome-wide as compared to WT mice (Fig. 2c). Fpg treatment, used to detect unrepaired base lesions, did not lead to increased comet tail lengths in any of the genotypes, suggesting no accumulation of base lesions in any of the mutants (Fig. 2c). This supports the mass spectrometry data showing unchanged global levels of oxidative DNA base lesions in all three mutants compared to WT (Fig. 2b). Third, a site-specific restriction enzyme-based qPCR method was applied to measure DNA lesions (i.e. apurinic/apyrimidinic (AP) sites, base lesions, strand breaks) or mutations in the *Gapdh* gene, which is transcriptionally active and thus, easily accessible for damage induction, but also repair. No significant changes were found, but as for the comet

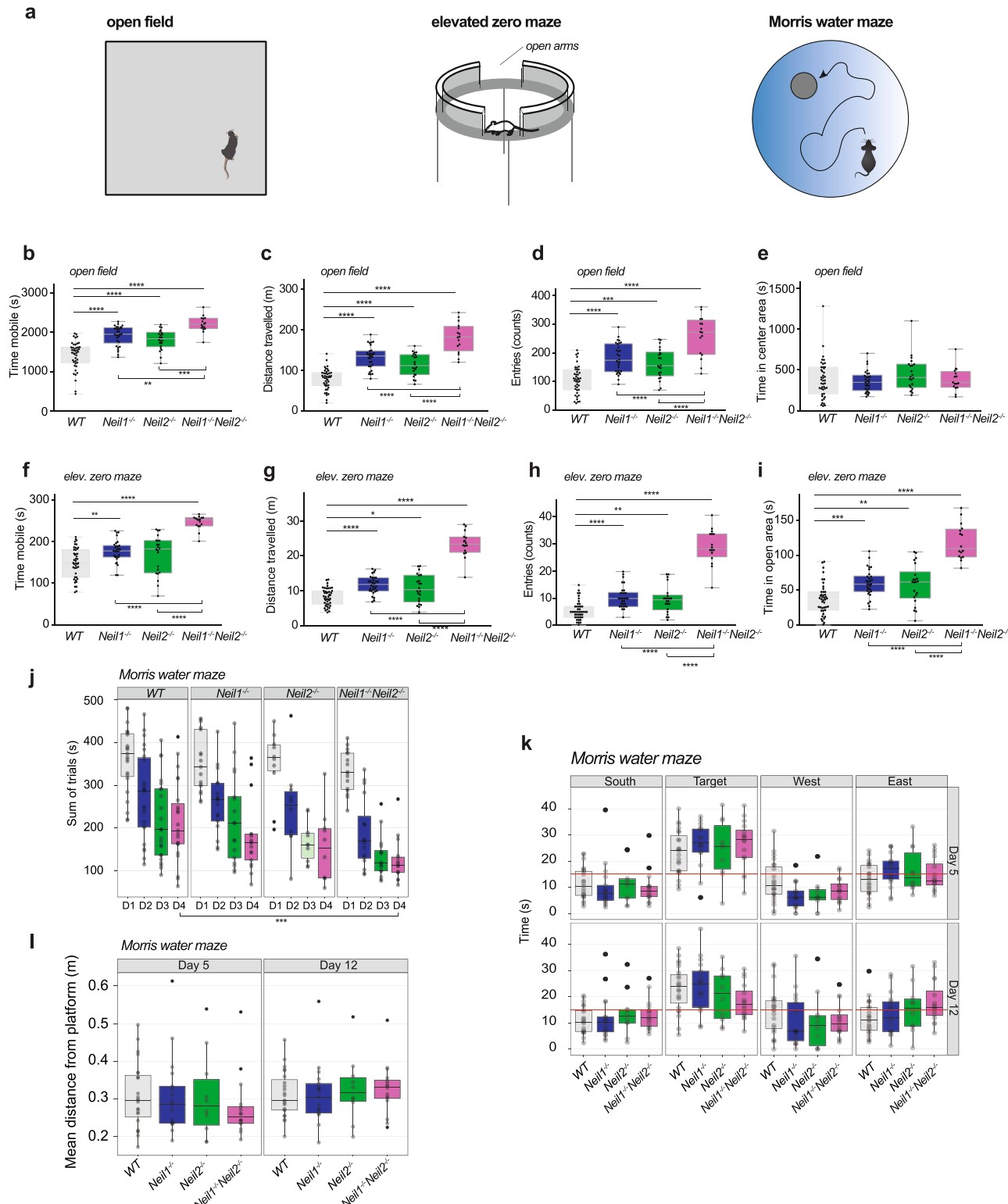

assay, there was a tendency to reduction in lesions/mutations in $Neil1^{-/-}Neil2^{-/-}$ hippocampi, as compared to WT ($P = 0.0856$) (Supplementary Fig. 2). In sum, our data suggest that the behavioral phenotype observed in $Neil1^{-/-}Neil2^{-/-}$ mice were not caused by impaired canonical BER.

Although the total steady-state levels of mutagenic oxidative DNA lesions were unaltered in hippocampi from adult mice lacking NEIL1 and/or NEIL2, mutations accumulated during development could still explain the phenotype at adulthood. To test this possibility, we applied whole-genome deep sequencing of hippocampal DNA from adult (6-month-old) male mice to determine mutation profiles. A DNA sequence variant analysis was performed using the WT hippocampus sample as the reference genome. We found a modest increase in DNA sequence variants genome-wide (Fig. 2d) that were evenly distributed across all chromosomes in all the three mutants (Fig. 2e). Variants were detected in all genomic regions with the majority occurring in non-coding regions, such as intergenic regions and

**Fig. 1 Increased activity, reduced anxiety and enhanced learning in Neil1$^{-/-}$Neil2$^{-/-}$ mice. a** Schematics of behavioral tests. **b–e** In the open field maze mice were allowed to explore freely for 45 min in an arena measuring L40 cm x W40 cm x H35 cm. An area of L20 cm x W20 cm was defined as the center area zone. **b** Time mobile, **c** distance travelled, **d** entries to the center area zone, and **e** time in the center area zone. **f–i** In the elevated zero maze, the mice were allowed 5 min for exploration on a 5-cm wide circular runway with alternating open and closed areas. **f** Time mobile, **g** distance travelled, **h** entries to the open area zones, and **i** time in the open area zones. **b–i** Data are shown in full, with overlaid boxplots representing the medians and the interquartile ranges (IQR). Whiskers indicate min/max values. Individual mice are represented by black dots. n = 43 WT, 30 Neil1$^{-/-}$, 22 Neil2$^{-/-}$ and 16 Neil1$^{-/-}$Neil2$^{-/-}$ mice. *P < 0.05, **P * p < 0.01, ***P < 0.001, ****P < 0.0001 by one-way ANOVA/Tukey. **j–l** In the Morris water maze, mice were trained to locate an escape platform hidden below the water surface (days 1–4) before memory was tested (days 5 and 12). **j** Latency to locate the platform and escape the water during learning trials, days 1 to 4. The data are shown as the total time spent in the tank for each mouse during eight trials. ***P = 0.00019 for Neil1$^{-/-}$Neil2$^{-/-}$ vs. WT by non-parametric pair wise Wilcoxon rank-sum test, Holm-adjusted. **k** Time spent in the four quadrants of the tank (the red line indicates average level expected by random behavior) and **l** mean distance from the platform zone during retention trials (probe tests), days 5 and 12. **j–l** Data are shown in full, with overlaid boxplots representing the medians and the interquartile ranges (IQR). Whiskers extend to a Tukey fence set at 1.5xIQR. Each mouse is indicated by a round symbol, outliers in black and all others in grey (different shades of grey indicate overlapping mice/symbols). n = 22 WT, 17 Neil1$^{-/-}$, 10 Neil2$^{-/-}$ and 16 Neil1$^{-/-}$Neil2$^{-/-}$ mice.

introns (Fig. 2f). Analysis of base-pair changes in SNPs showed a normal distribution with C:G to T:A transitions being the most frequent, most likely due to deamination of 5mC and C to thymine and uracil, respectively (Fig. 2g). These results indicate that lack of NEIL1 and/or NEIL2 does not lead to a genome-wide hypermutator phenotype in the hippocampus.

**Reduced axonal activation in stratum oriens of Neil1$^{-/-}$Neil2$^{-/-}$ hippocampus.** To assess potential changes in excitatory synaptic transmission and cell excitability that could possibly explain the altered behavior in NEIL1/NEIL2-deficient mice, we recorded in either stratum radiatum (SR) or stratum oriens (SO) and simultaneously in stratum pyramidale (SP) in the CA1 region of hippocampal slices from adult (4-month-old) male Neil1$^{-/-}$Neil2$^{-/-}$ and WT mice. We decided to focus on the hippocampal CA1 subfield due to its prominent role in both spatial information coding and anxiety regulation[11,12,35,36]. The stimulation intensities necessary to elicit prevolleys of given amplitudes (0.5, 1.0, and 1.5 mV) tended to be higher, though not statistically significant, in SR of Neil1$^{-/-}$Neil2$^{-/-}$ mice compared to WT mice (Fig. 3a). Similar tendencies were observed in SO, and for one of the prevolley amplitudes (0.5 mV), the difference between mutant and control mice reached statistical significance (Fig. 3f). Measuring the field excitatory postsynaptic potential (fEPSP) as a function of the same prevolley amplitudes showed that Neil1$^{-/-}$Neil2$^{-/-}$ animals evoked fEPSPs similar to those obtained in WT mice, in both SR (Fig. 3b, e) and SO (Fig. 3g, j). Furthermore, postsynaptic excitability, measured as fEPSPs necessary for generating a population spike, was not significantly changed in Neil1$^{-/-}$Neil2$^{-/-}$ mice compared to WT mice in SR (Fig. 3c, e) or SO (Fig. 3h, j). In sum, the results do not support any major differences between the mutant and WT mice in excitatory synaptic transmission (Fig. 3b, g) or postsynaptic excitability (Fig. 3c, h) in either of the two strata examined. However, in SO, slightly altered axonal activation (Fig. 3f) could indicate a reduction in fiber density, number of afferent fibers or a differential receptor composition in receptor subunits in Neil1$^{-/-}$Neil2$^{-/-}$ mice compared to WT.

To further characterize excitatory synaptic transmission in the hippocampal CA1 region, we measured paired-pulse facilitation (PPF)[37], a short-lasting form of synaptic plasticity primarily attributed to changes in presynaptic $Ca^{2+}$ homeostasis[38]. A comparison of PPF did not reveal any differences between the two genotypes in SR (Fig. 3d) or in SO (Fig. 3i).

We next analyzed the long-term potentiation of synaptic transmission (LTP) at CA3 to CA1 synapses in WT and Neil1$^{-/-}$Neil2$^{-/-}$ mice in SR and SO. Tetanic stimulation of the afferent fibers in either of the pathways produced a lasting, homosynaptic potentiation of the fEPSP slope of similar magnitude in Neil1$^{-/-}$Neil2$^{-/-}$ and control mice, when measured 40 – 45 min after the tetanizations (Fig. 3k, l). In both SR and SO, LTP in Neil1$^{-/-}$Neil2$^{-/-}$ mice was similar in magnitude to the corresponding pathways in WT mice.

**NEIL1 and NEIL2 differentially affect gene expression in CA1 with a potential relevance for synaptic function, plasticity, and composition.** We recently reported hippocampal transcriptional changes in mice lacking OGG1 and MUTYH DNA glycosylases[16]. We therefore asked whether NEIL1 and NEIL2 DNA glycosylases could similarly act as transcriptional regulators within the hippocampus to modulate synaptic transmission and behavior. As for the electrophysiology, we focused on the CA1 subfield of the hippocampal formation due to its role in spatial learning and anxiety[11,12,35,36]. We applied whole-genome sequencing of RNA isolated from the pyramidal layer of CA1 of adult (3–6-month-old) male mice by laser capture microdissection (Fig. 4a), a method, which offers supreme tissue specificity[39]. A moderate number of differentially expressed genes (DEGs) were detected, the largest amount in Neil2$^{-/-}$ and Neil1$^{-/-}$Neil2$^{-/-}$ mice (Fig. 4b, Supplementary Fig. 3d–f). Notably, there were no significant changes in the expression of other oxidative DNA glycosylases, such as Ogg1, Neil3 and Nth1 (Supplementary Fig. 3d–f), indicating that there is no compensatory upregulation of these repair genes in mice lacking NEIL1 and/or NEIL2. Similar numbers of up- and downregulated genes were found in all genotypes (Fig. 4c). While there was almost no overlap in DEGs between Neil1$^{-/-}$ and Neil2$^{-/-}$, we found most overlapping DEGs between single- and double-knockout mice (Fig. 4d). Exploratory data analysis identified two samples to be clear outliers (Supplementary Figs. 3g and 4). These two outliers (red arrows, Supplementary Fig. 3g) were excluded from the group comparison analysis. A reactome-pathway analysis showed the nuclear receptor signaling pathway (R-MMU-38328) to be significantly overrepresented in Neil1$^{-/-}$Neil2$^{-/-}$ CA1. Of note, all three isotypes of the orphan nuclear receptor Nr4a were downregulated in Neil1$^{-/-}$Neil2$^{-/-}$ mice, whereas the nuclear receptors Nr1d1 and Nr1d2 were upregulated (Fig. 4e). While four of these five nuclear receptors were similarly differentially regulated in Neil2$^{-/-}$, none of them were altered in Neil1$^{-/-}$ CA1, pointing to a NEIL2-dependent regulation of nuclear receptors. The top10 downregulated genes in Neil1$^{-/-}$Neil2$^{-/-}$ mice largely overlapped with those of Neil1$^{-/-}$ mice, whereas upregulated genes overlapped mainly with those of Neil2$^{-/-}$ mice (Fig. 4f). Among the up- and downregulated DEGs we identified four genes as immediately relevant to synaptic function according to their QuickGO annotation[40] (Fig. 4f). While three of them (Npbwr1, Htr3a and Fxyd2) play a role in a very specific subset of receptor systems and synaptic membrane elements, Npas4 is a well-characterized master regulator of inhibitory synapse development[41]. The latter was differentially regulated distinctly in Neil1$^{-/-}$ and Neil1$^{-/-}$Neil2$^{-/-}$

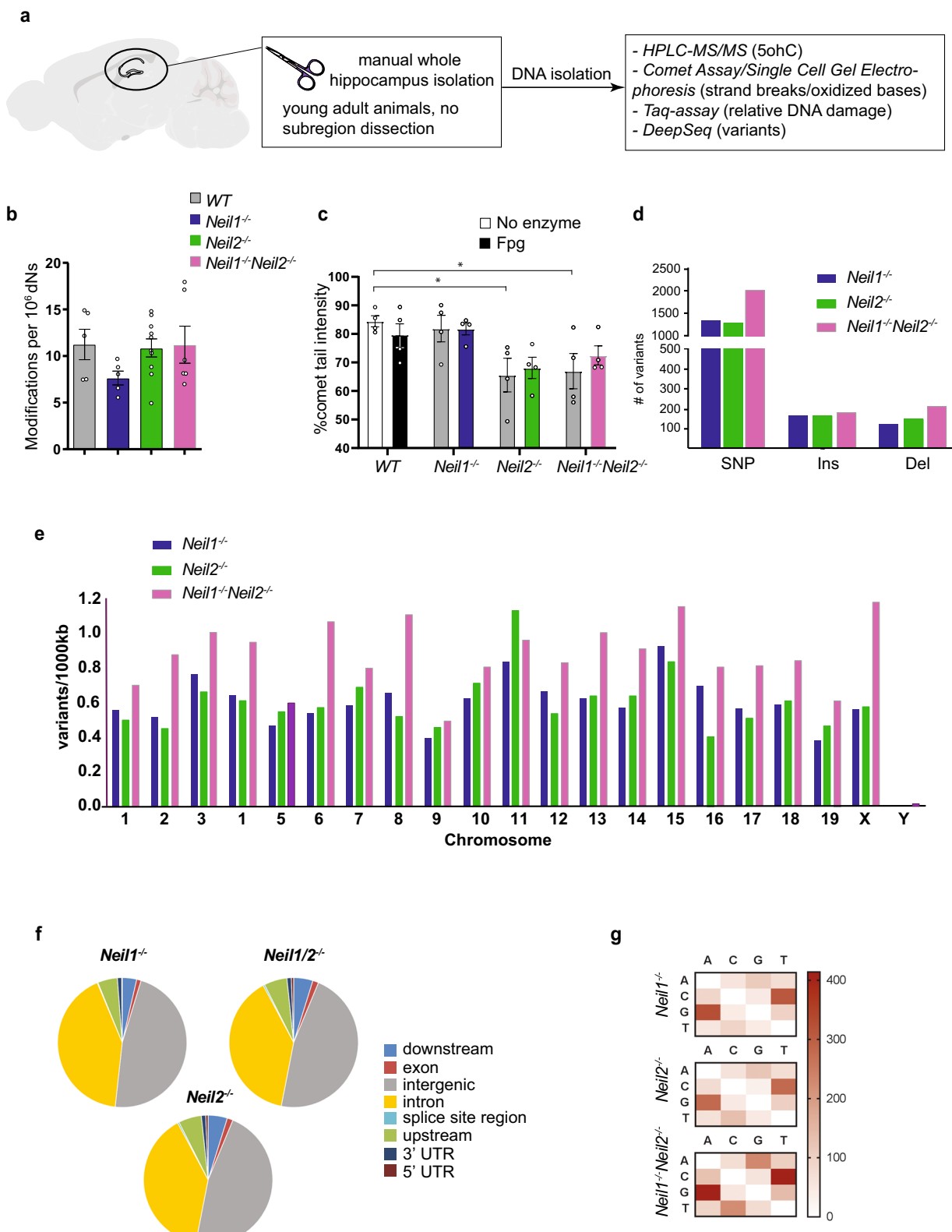

animals, but not in $Neil2^{-/-}$ animals, indicating a mainly NEIL1-dependent regulation of this gene (Fig. 4f). Differences observed in group comparisons were also visible at a single animal level (Fig. 4g). To further thematically cluster the DEGs found in the different genotypes, we performed a gene ontology biological processes enrichment analysis (PANTHER release 2020-07-28, GO database release 2020-07-16, DEGs log2fold(abs) >0.3, p < 0.05). All

NEIL-deficient mice showed an enrichment of several GO-terms immediately relevant to central nervous system function (colored GO-terms, Supplementary Fig. 3a–c), further highlighting the relevance of NEIL DNA glycosylases in CA1 transcription regulation. Interestingly, DEGs of $Neil1^{-/-}$, $Neil2^{-/-}$ and $Neil1^{-/-}Neil2^{-/-}$ mice showed enrichment for high-expressed genes (48%, 41% and 45%, respectively), as represented by the number of DEGs

**Fig. 2 Unchanged steady-state levels of oxidative DNA base lesions and no hypermutator phenotype in *Neil1*<sup>−/−</sup>*Neil2*<sup>−/−</sup> hippocampus.**
**a** Hippocampus was isolated from WT and NEIL-deficient mice and DNA damage and mutation levels estimated by various methods. **b** HPLC-MS/MS analysis of 5-ohC in hippocampal, genomic DNA. Data are shown as mean ± SEM. (individual mice are represented by circles). $n = 5$ WT, 5 *Neil1*$^{-/-}$, 10 *Neil2*$^{-/-}$ and 6 *Neil1*$^{-/-}$*Neil2*$^{-/-}$ mice. **c** DNA damage levels in hippocampal tissue by alkaline comet assay analysis. Data are shown as mean of gel medians (circles indicate gel median for each mouse; 50 comets x 3 gels scored per mouse) ± SEM. $n = 4$ mice per genotype. *$P = 0.0291$ and 0.0495 for *Neil2*$^{-/-}$ and *Neil1*$^{-/-}$*Neil2*$^{-/-}$ vs. WT, respectively, by two-way ANOVA/Sidak. **d–g** DNA samples from four mice of each genotype were pooled and subjected to whole-genome deep sequencing followed by mutation profile analysis (for details, see Methods). **d** DNA sequence variants, **e** Chromosomal distribution of DNA sequence variants, **f** Genomic region distribution of DNA sequence variants, and **g** Base changes count of SNPs in NEIL-deficient vs. WT hippocampus. SNP, single nucleotide polymorphism; Ins, insertions and Del, deletions.

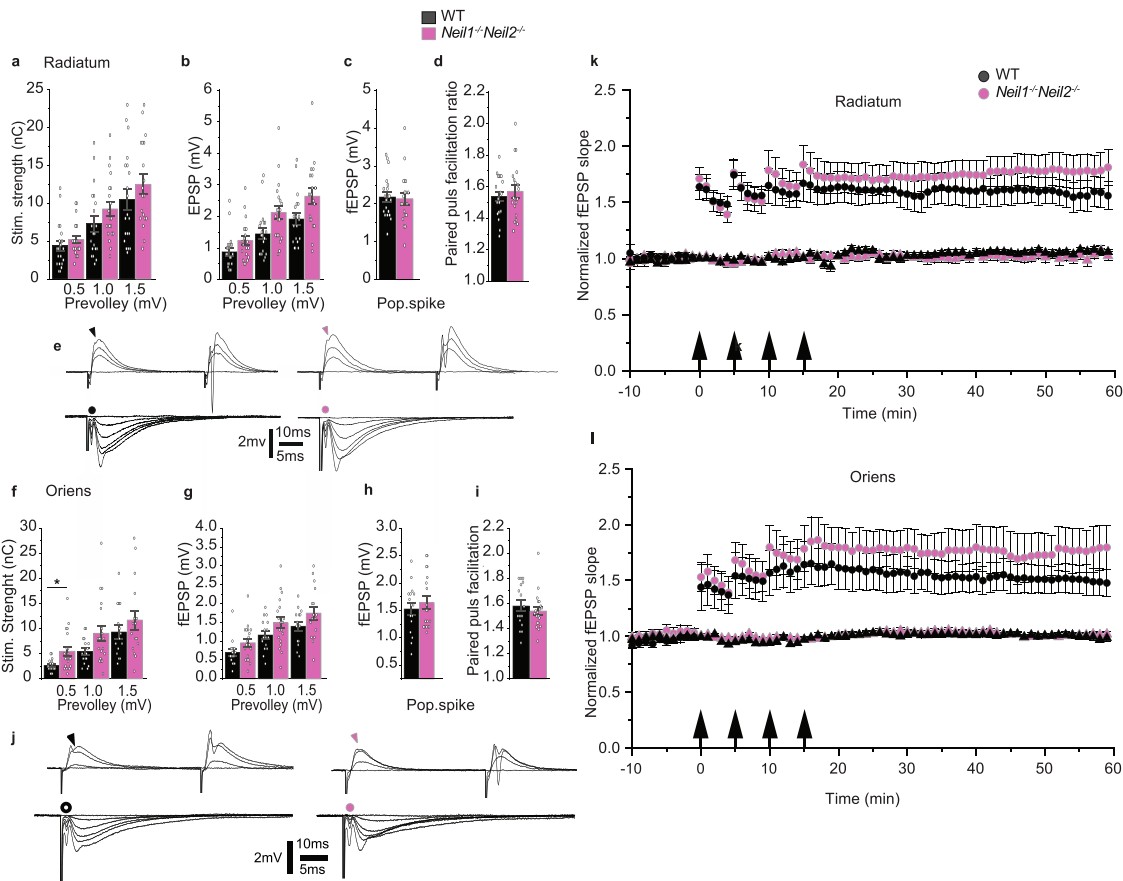

**Fig. 3 Reduced axonal activation in stratum oriens of *Neil1*<sup>−/−</sup>*Neil2*<sup>−/−</sup> hippocampus. a–j** Synaptic transmission, excitability and paired-pulse facilitation (PPF) in stratum radiatum (SR; **a–e**) and stratum oriens (SO; **f–j**) of *Neil1*$^{-/-}$*Neil2*$^{-/-}$ and WT mice. **a, f** Stimulation strengths necessary to elicit prevolleys of given amplitudes (0.5, 1.0, and 1.5 mV). **b, g** fEPSP amplitudes as a function of the same three prevolley amplitudes. **c, h** The fEPSP amplitudes necessary to elicit a just detectable population spike. **d, i** PPF ratio from the two genotypes at an interstimulus interval of 50 ms. **e, j** Recordings from stratum pyramidale (SP) elicited by paired-pulse stimulation (50 ms interstimulus interval). Arrowheads indicate the population spike thresholds in control (black) and *Neil1*$^{-/-}$*Neil2*$^{-/-}$ (magenta) mice. Circles indicate prevolleys preceding fEPSPs in control (black) and *Neil1*$^{-/-}$*Neil2*$^{-/-}$ (magenta) mice. Each trace is the mean of five consecutive synaptic responses elicited by different stimulation strengths. **a–d, f–i** Data are shown as mean ± SEM and *n* values represent total hippocampal slices per genotype (4–6 per mouse). Measures in individual slices are indicated with circles superimposed on the mean bars. **a, b** $n = 20$ WT for each of the stimulation strengths and 20, 20, 19 *Neil1*$^{-/-}$*Neil2*$^{-/-}$ for 0.5, 1.0, and 1.5 mV, respectively. **c** $n = 20$ WT and 19 *Neil1*$^{-/-}$*Neil2*$^{-/-}$. **d** $n = 20$ for both genotypes. **f, g** $n = 16, 16, 15$ WT and 20, 20, 16 *Neil1*$^{-/-}$*Neil2*$^{-/-}$ for 0.5, 1.0, and 1.5 mV, respectively. **h** $n = 16$ WT and 19 *Neil1*$^{-/-}$*Neil2*$^{-/-}$. **i** $n = 16$ WT and 20 *Neil1*$^{-/-}$*Neil2*$^{-/-}$. **f** *$P = 0.019$ for prevolley of 0.5 mV, by linear mixed model analysis. **k, l** Normalized and pooled fEPSP slopes evoked in hippocampal slices from WT and *Neil1*$^{-/-}$*Neil2*$^{-/-}$ mice in SR (**k**) and SO (**l**). The tetanized pathways are shown as circles and the untetanized control pathways are shown as triangles. Arrows indicate time points of tetanic stimulation. Data are shown as mean ± SEM and n-values represent total hippocampal slices per genotype (4–6 per mouse). **k** $n = 10$ WT and 11 *Neil1*$^{-/-}$*Neil2*$^{-/-}$ **l** $n = 9$ WT and 8 *Neil1*$^{-/-}$*Neil2*$^{-/-}$.

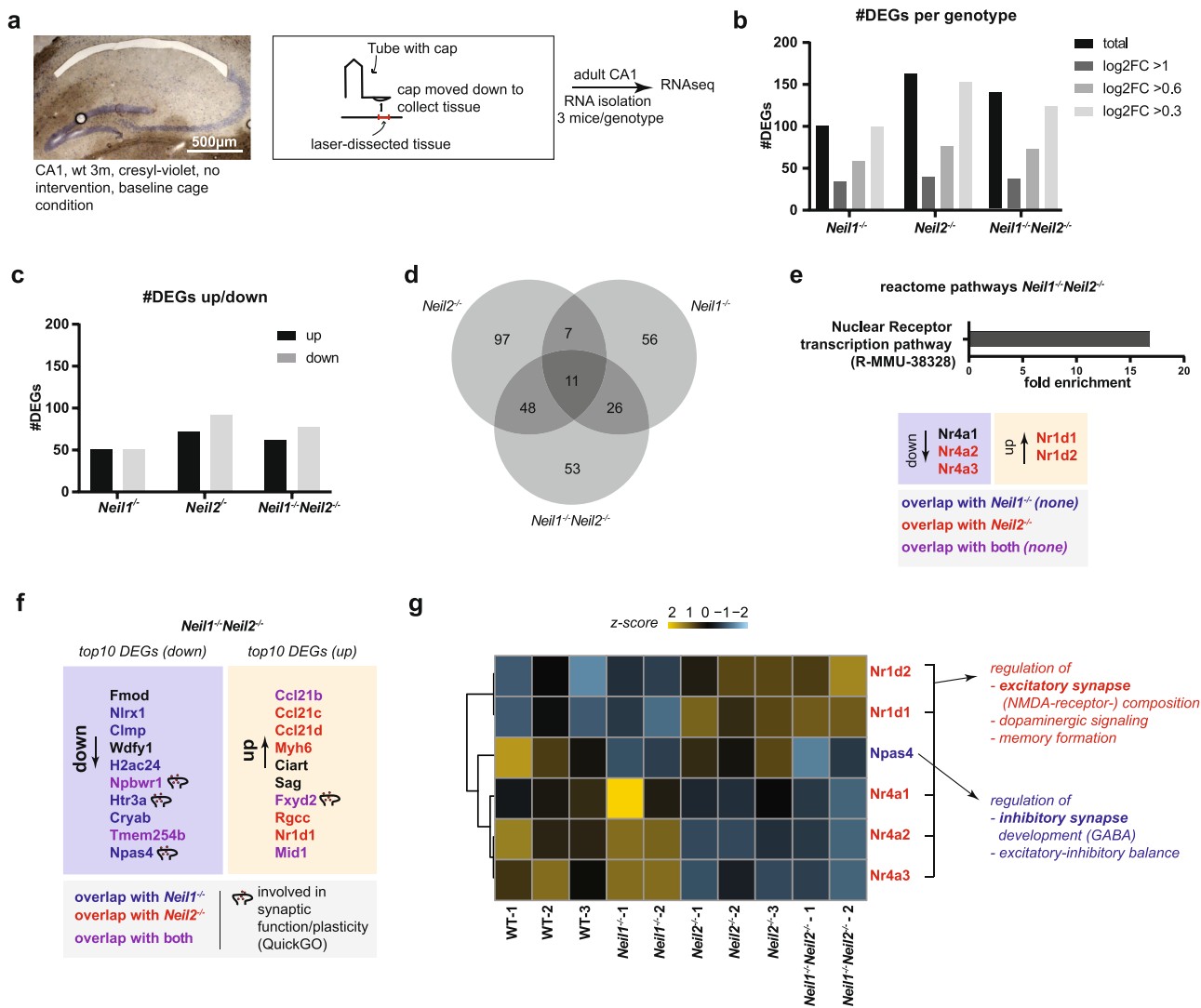

**Fig. 4 NEIL1 and NEIL2 jointly affect the CA1 transcriptome. a** Tissue used for RNAseq was isolated using a laser dissection approach (see Methods for details). **b** Amount of DEGs found for each genotype (based on pooled data from three (WT and $Neil2^{-/-}$) or two ($Neil1^{-/-}$ and $Neil1^{-/-}Neil2^{-/-}$) mice, see also Supplementary Fig. 3). DEGs above each log2fold cut-off are displayed in different shades of grey. **c** Up- and downregulated genes for each mutant. **d** Overlapping DEGs between single and double knockouts. **e** Enriched reactome-pathway, R-MMU-38328 (fold enrichment 19.72, FDR 1.19e-02), and corresponding DEGs in $Neil1^{-/-}Neil2^{-/-}$ mice. Overlap with $Neil1^{-/-}$ or $Neil2^{-/-}$ mice is indicated. **f** top10 down- and upregulated DEGs in $Neil1^{-/-}Neil2^{-/-}$ mice. (Npas4: −1.12 log2fold). Overlap with $Neil1^{-/-}$ or $Neil2^{-/-}$ mice is indicated. **g** Expression of relevant DEGs at a single animal level (z-score table based on FPKMs). Thematic relevance of DEGs in $Neil1^{-/-}Neil2^{-/-}$ mice is shown to the right.

in the 75% upper quartile of all sequenced genes. In contrast, less than 1.5% of DEGs in the mutant mice were low-expressed genes as demonstrated by the number of DEGs in the 25% lower quartile of all sequenced genes (Supplementary Fig. 5).

**Altered synaptic composition in $Neil2^{-/-}$ and $Neil1^{-/-}Neil2^{-/-}$ mice.** Based on the transcriptome results showing differential regulation of factors relevant for synaptic composition, we decided to examine the excitatory and inhibitory transmitter systems within the CA1 subregion of the hippocampal formation by immunohistochemistry. We chose to study the NMDA- and GABA-receptors due to their reciprocal interaction with both Npas4 and Nr4a-isoforms and their previously shown association with NEIL-deficiency[14]. As a first approximation, we picked an immunohistochemistry-approach looking specifically into the CA1 subregion of the hippocampus of 3–6-month-old male mice (see also Methods and Discussion).

Within the tetrameric structure of the NMDA-receptor complex (Supplementary Fig. 6, illustration), regulatory subunits such as NR2A (GRIN2A) and NR2B (GRIN2B) determine the receptor's electrophysiological properties and are seen as important mediators of synaptic plasticity[42]. We therefore primarily examined these two subunits of the NMDA receptor. Across SP, we found significantly reduced NR2A-reactivity in $Neil2^{-/-}$ and $Neil1^{-/-}Neil2^{-/-}$ mice compared to WT (Supplementary Fig. 6a). NR2A reactivity was also significantly lower within SO in $Neil1^{-/-}Neil2^{-/-}$ mice compared to WT (Supplementary Fig. 6a). As for the NR2B subunit, reduced reactivity was observed across SP of $Neil2^{-/-}$ mice only (Supplementary Fig. 6b). A low NR2A/NR2B-ratio has previously been reported to enhance both memory formation[43] and LTP[44]. We observed a significantly reduced NR2A/NR2B ratio exclusively in $Neil1^{-/-}Neil2^{-/-}$ mice, with the most prominent reduction across SO (SO, Δ2.558; SR, Δ1.685; SP, Δ1.053) (Supplementary Fig. 6c), the region that showed significantly reduced axonal activation (Fig. 3f).

Npas4 has been shown to coordinate inhibitory signaling via the GABA-A-receptor, both in vitro and in vivo[45,46]. We chose to examine specifically the GABA-A-receptor alpha2 subunit (GABRA2) as it is involved in anxiety regulation via distinct intrahippocampal circuits[47]. As for NR2A (Supplementary Fig. 6a), the GABRA2-reactivity was significantly reduced in $Neil1^{-/-}Neil2^{-/-}$ mice compared to WT mice across SP (Supplementary Fig. 7a). In the $Neil1^{-/-}$ and $Neil2^{-/-}$ mice we observed a tendency to reduction in SP; however, this was not statistically significant (Supplementary Fig. 7a). While the expression of NR2A across SO and SR of $Neil2^{-/-}$ and $Neil1^{-/-}Neil2^{-/-}$ mice showed a similar tendency to reduction as in SP (Supplementary Fig. 6a), the differences were less conclusive for GABRA2 in SO and SR (Supplementary Fig. 7a).

Next, we examined the expression of Postsynaptic density-95 (PSD-95), an abundant postsynaptic scaffolding protein associated with the NMDA-receptor complex[48]. In line with reduced absolute levels of NR2A in SP of $Neil2^{-/-}$ and $Neil1^{-/-}Neil2^{-/-}$ mice (Supplementary Fig. 6a) and NR2B in SP of $Neil2^{-/-}$ mice (Supplementary Fig. 6b), we found a tendency to reduced PSD-95 immunoreactivity in SP in both mutants (P = 0.0762 and 0.0983, respectively) (Supplementary Fig. 7b).

In sum, these results could point to a potential instability of NMDA-receptor architecture within the postsynaptic compartment in the context of NEIL1/NEIL2 deficiency (for limitations of this approach, see discussion).

## Discussion
The current study revealed an altered behavioral phenotype in mice deficient in both the NEIL1 and NEIL2 DNA glycosylases, shown as increased locomotor activity in the OF test and the EZM, reduced anxiety in the EZM, and improved learning ability in the MWM test. We have previously reported similar observations in $Ogg1^{-/-}Mutyh^{-/-}$ mice. However, in the $Ogg1^{-/-}Mutyh^{-/-}$ mice, learning was impaired[16]. Further, we recently demonstrated that mice carrying one deficient allele of $Ogg1$ exhibited poorer early-phase learning performance than WT mice using the Barnes maze, and that it was restored when the mice were subjected to oxidative stress by X-ray irradiation[49]. Inactivation of NEIL3 DNA glycosylase induced an anxiolytic effect and a tendency to impaired learning in mice, however, without increased locomotor activity[14]. In contrast, over-expression of the repair gene hMTH1, preventing 8-oxoG accumulation in the brain, also reduced anxiety in mice without an increase in activity level[50]. Thus, it appears that DNA glycosylases affect processes involved in behavior and cognition in distinct ways. Canugovi and coworkers previously reported similar learning ability, but defects in short-term spatial memory retention in NEIL1-deficient mice[23]. Correspondingly, no learning defects were observed in our NEIL1-deficient mice; however, memory was not affected either. A possible explanation to this discrepancy could be that the mice used in the present study were younger (6 months) than the mice tested by Canugovi and colleagues (9–33 months). Further, we have previously shown that $Neil1$ mRNA expression increases with age in mouse brains[29], suggesting that NEIL1 could be important for cognitive functions at a later stage than we have explored here. It may be argued that comparing mutant mice to control mice that are not littermates, could affect the outcome of the behavioral tests. However, since all lines were backcrossed onto the same background as control mice (C57BL/6 N), breeding of separate lines should not affect the results notably. The increased weight observed in $Neil1^{-/-}$ mice, but not in $Neil2^{-/-}$ and $Neil1^{-/-}Neil2^{-/-}$ mice, could indicate that inactivation of $Neil2$ rescues the weight phenotype observed in the $Neil1^{-/-}$ mice, or it could be due to variable penetrance of

the metabolic phenotype observed in the $Neil1^{-/-}$ mice[32]. Since $Neil1^{-/-}$ and $Neil2^{-/-}$ mice show similar behavior, regardless of the weight differences, metabolic function and health status in $Neil1^{-/-}$ mice is unlikely to have an impact on behavior.

NEIL DNA glycosylases are assumed to be important for genome maintenance by preventing the accumulation of oxidative DNA damage. It is therefore reasonable to expect increased levels of oxidative base lesions and possibly mutations when these enzymes are lacking. In line with this, elevated levels of FapyA lesions, but not FapyG or 8-oxoG, were detected in brains from adult (9–22 months) NEIL1 KO mice[51]. NEIL2 KO mice have also been shown to accumulate oxidized DNA bases in various organs, including the brain, but mainly in transcribed regions[33]. In the present study, accumulation of hippocampal DNA damage was not detected in any of the DNA glycosylase-deficient strains studied and RNA sequencing analysis did not reveal any compensatory upregulation of other DNA glycosylases in CA1 of the hippocampus. Further, only a modest increase in DNA variants in NEIL-deficient hippocampi was found. Although a slightly higher number of variants were detected in the double mutant compared to the single mutants, the number is too small (< 2500 per genome) for the double mutant to be characterized as a hypermutator. Similar observations were made in the spleen, liver, and kidney of NEIL1/NEIL2-deficient mice, which showed neither increased mutation frequencies nor cancer predisposition under normal physiology[34]. Further, no global increase in 8-oxoG levels was detected in the hippocampus or hypothalamus of mice deficient in both the OGG1 and MUTYH DNA glycosylases[16]. Thus, impaired or reduced global (canonical) repair of oxidized DNA bases in brain regions involved in cognition is not likely to explain the altered behavioral phenotypes observed in DNA glycosylase-deficient mice. Intriguingly, $Neil2^{-/-}$ and $Neil1^{-/-}Neil2^{-/-}$ mice showed reduced levels of DNA damage in the hippocampus. If NEIL2 plays a role in chromatin modulation, the NEIL2-deficient mice may contain more heterochromatin. Thus, we may speculate that reduced DNA damage is caused by a putative role of NEIL2 in processes making the chromatin more accessible to strand breaks.

We recently reported that transcriptional changes in the hippocampus of mice lacking OGG1 and MUTYH DNA glycosylases could be an underlying cause of the altered behavioral phenotype observed[16]. Further, in $Ogg1^{+/-}$ hippocampus, the expression of three of 35 genes investigated was correlated to spatial learning in the Barnes maze[49]. Thus, to begin to elucidate the mechanisms behind the behavioral alterations observed in the present study, we looked for changes in the hippocampal CA1 transcriptome. NEIL DNA glycosylases have previously been suggested to be involved in gene regulation by repairing preferentially transcribed genes and quadruplex DNA in promoter regions[33,52]. In support of this, our RNA sequencing followed by transcriptome analysis revealed that in particular, highly expressed genes show differentially expression upon loss of NEIL DNA glycosylases. Notably, DEGs within the CA1 pyramidal layer of NEIL1/NEIL2-deficient mice referred to genes highly relevant for behavior, synaptic composition and function. Loss of NEIL2 appears to specifically affect Nr4a orphan receptors, with all three isoforms down-regulated in $Neil1^{-/-}Neil2^{-/-}$ mice, and largely overlapping with $Neil2^{-/-}$ mice. Consequently, DEGs in $Neil1^{-/-}Neil2^{-/-}$ were significantly enriched in the nuclear receptor signaling pathway. In CA1, the nuclear receptor signaling pathway is particularly important for regulating excitatory synapse composition[53], dopaminergic signaling and, in general, processes of memory formation[54]. Nr4a1 (Nur77), whose function is enhanced when it forms heterodimers with Nr4a2 (Nurr1)[55], interacts reciprocally with (excitatory) NMDA-receptor signaling. It regulates spine density and excitatory synapse distribution, especially at distal

dendritic compartments[53]. Further, reduced expression of Nr4a2 has previously been linked to a hyperactive behavior phenotype in mice[56,57]. This indicates a mechanistically relevant impact of NEIL1 and NEIL2 on these receptors to modulate adaptive behavior. As another example of NEIL1 and NEIL2 interacting with gene expression relevant for synaptic composition and function, we observed Npas4 to be downregulated both in $Neil1^{-/-}Neil2^{-/-}$ and $Neil1^{-/-}$ animals. Npas4 is prominently involved in regulating the excitatory-inhibitory balance within neural circuits[41], with a particular relevance for GABAergic (inhibitory) signaling[45]. In sum, this suggests that NEIL1 and NEIL2 glycosylases jointly affect the expression of genes relevant for synaptic composition and function, with NEIL2 being prominently involved in nuclear receptor signaling and NEIL1 mainly involved in Npas4-regulation.

With Npas4 being a regulator in excitatory-inhibitory balance and Nr4a receptors interacting directly with the NMDA-receptor, we further examined the expression of the NMDA-receptor in the context of NEIL-deficiency by immunohistochemistry. Here, we focused on the regulatory subunits NR2A and NR2B due to their eminent role in determining the receptor's electrophysiological properties as well as its relevance in behavior and pathophysiology[42]. The results we present here point to a potentially reduced NR2A/NR2B ratio in NEIL1/NEIL2-deficient mice compared to WT, which constitutes a further refinement of previously reported altered NMDA-receptor composition[14]. This may partly explain the behavioral phenotype of improved spatial learning in NEIL1/NEIL2-deficient mice, since recent findings suggest a low NR2A/NR2B ratio to be associated with improved memory acquisition performance[43] and enhanced LTP[44]. However, we need to point out that our approach bears certain limitations requiring further studies to confirm the observations we report here: While we chose a highly subregion-specific approach with a fluorescent immunosignal-based quantification selectively within the regions of interest (see Methods), we cannot present subregion-specific data at a protein level confirming these results. Thus, we chose to present the immunohistochemistry data as part of the Supplementary Material (Supplementary Fig. 6). Confirmation of results could be done e.g. with a mass spectrometry approach looking specifically into CA1-SP laser dissectates; however, this is beyond the scope of this study.

The coupling between LTP in CA1 and spatial reference memory in the MWM has been questioned in recent studies[11,58,59]. Thus, the unaltered LTP in the $Neil1^{-/-}Neil2^{-/-}$ mice does not necessarily contradict the improved learning. While we did not observe differences in LTP, $Neil1^{-/-}Neil2^{-/-}$ mice displayed electrophysiological changes in form of decreased axonal activation in SO of the hippocampal CA1 subregion. However, these differences were observed for one specific pre-volley amplitude only. Nonetheless, we think that this may point to a reduced number of afferent fibers in SO. Interestingly, recent evidence suggests that the inhibition of heterogeneously tuned excitatory afferent input to CA1 is beneficial for spatial coding[60]. One could therefore speculate that a decrease in afferent fiber density may cause reduced excitatory input to CA1 in the context of NEIL1/NEIL2 deficiency, sufficient for improved spatial coding, at least in the very general spatial learning context of a MWM. However, spatial information coding is distinctly a network task involving all hippocampal subfields as well as the entorhinal cortex[61]. Our study only examines the, albeit very important, CA1 subfield in detail and the behavioral read-out used for this study do not permit conclusions about more refined elements of spatial coding such as pattern completion.

Throughout our study, we used robust and strict statistical approaches and we do think that our experiments were sufficiently powered to draw the conclusions we present in this manuscript (see considerations on a priori/post-hoc power analyses in Methods). Yet, our RNAseq approach relied on a relatively small sample size ($n = 3$) due to the technically challenging, time- and cost-intensive experimental approach. The statistical approach we used to analyze our RNAseq data takes this into account (DESeq2, see Methods), thereby allowing to draw conclusions with a reasonable reliability. We recommend nonetheless that future studies looking into the behavioral, electrophysiological and biomolecular description of new transgenic mouse models in DNA repair anticipate a high data variance, as observed e. g. in Fig. 1, and take this into account in a priori power calculations.

Recently, NEIL1 was identified as a potential reader of oxidized cytosine derivatives, and both NEIL1 and NEIL2 were suggested to potentially cause gene reactivation by an alternative BER pathway for DNA methylation[62,63]. Furthermore, both proteins were shown to promote substrate turnover by TDG (Thymine-DNA glycosylase) during DNA demethylation[64]. This suggests a role in gene regulation, possibly involving epigenetics[64,65]. In light of this, the behavioral phenotype observed in NEIL1/NEIL2-deficient mice does not seem to be caused by impaired canonical repair of oxidative base lesions. Instead, our results point to a NEIL1/NEIL2-dependent regulation of synaptic factors both at RNA and protein level that is not explained by the enzymes' function in DNA repair, but rather their noncanonical contribution to gene regulation.

## Methods

**Experimental model and subject details.** All experiments were approved by the Norwegian Animal Research Authority and conducted in accordance with the laws and regulations controlling experimental procedures in live animals in Norway and the European Union's Directive 2010/63/EU. NEIL1 KO ($Neil1^{-/-}$), NEIL2 KO ($Neil2^{-/-}$) and NEIL1/NEIL2 DKO ($Neil1^{-/-}Neil2^{-/-}$) mouse models generated previously in our lab[34], were used throughout the study. The three mutant lines, all backcrossed for at least eight generations onto the C57BL/6 N background, were bred separately to obtain mice for experiments and C57BL/6 N mice were included as WT controls. Of note, breeding of separate lines reduces the number of mice needed for experiments and is in agreement with general practice, as long as the mice are backcrossed onto the same background as control mice (https://www.jax.org/jax-mice-and-services/customer-support/technical-support/breeding-and-husbandry-support/considerations-for-choosing-controls). The mice were housed and bred in a 12-h-light/dark cycle at the Department of Comparative Medicine, Oslo University Hospital, Rikshospitalet, Norway, or the Comparative Medicine Core Facility, NTNU, Trondheim, Norway, with food and water *ad libitum*. The mice were housed with their littermates (max five in each cage). Throughout the study we have used male mice aged 3 to 6 months. This age group is usually referred to as mature adult and consists of mice that are past development, but not yet affected by senescence[66]. It is a relatively homogeneous group when it comes to the parameters we have investigated in the present study[67,68]. Specific age is stated in respective methods and results sections. Different cohorts of mice were used in each experiment, except for in behavioral studies where some of the mice were used in all three tests.

**Behavioral studies.** Behavioral studies were performed on 6-month-old mice[16]. The mice were subjected to the behavioral tests in the following order: Open Field Maze (OFM), Elevated Zero Maze (EZM) and Morris Water Maze (MWM). The same mice were used in all three tests, however, in the OF and EZM more mice were included, as these tests are less time consuming than the MWM. The mice subjected to the MWM were weighed after the last probe test. The OF test[69] monitoring general locomotor activity was conducted in an arena measuring L40 cm x W40 cm x H35 cm, where the middle of the arena, L20 cm x W20 cm, was defined as the center area zone. Mice were allowed to explore freely for 45 min. The EZM task[70] measuring activity and anxiety, was conducted on a circular runway 60 cm above the floor with four alternating open and closed areas. The mice were placed on the maze facing a closed area and allowed 5 min for exploration of the apparatus. An open area entry was defined as 85% of the mouse being inside an open area. Learning and memory were monitored using the MWM[71]. Testing was carried out in a white circular pool, 120 cm in diameter and filled 2/3 with white, opaque water (SikaLatex liquid, Sika, Norway) kept at 22 ± 1 °C. Using visual cues, the mice learned to find a hidden escape platform, 11 cm in diameter and located at a fixed position 0.5–1.0 cm below the water surface, during repeated daily sessions (days 1–4). The mice were released in the water facing the wall of the pool at four fixed positions in a pseudorandom sequence and given a maximum of 60 s to locate the hidden platform. Each mouse had eight trials each day in the training period,

four in the morning and four in the afternoon. After each block (four trials) the mouse was placed in a heated cage to dry before being returned to the home cage. On days 5 and 12, each mouse was subjected to a single retention trial of 60 sec (probe test) to test spatial memory. During retention trials, the escape platform was submerged to the bottom of the pool. A spatial bias for the target quadrant constitutes evidence for spatial memory. During all three behavioral tests, positions of the mice were tracked and stored by using ANY-maze video tracking system (Stoelting, IL, USA).

**DNA damage analysis**. *HPLC-MS/MS analysis.* DNA was isolated from hippocampi of 4–6-month-old WT and NEIL-deficient male mice using DNeasy Blood and Tissue kit (Qiagen, cat. no. 80004), according to manufacturer's protocol. Two µg of genomic DNA was enzymatically hydrolyzed to deoxyribonucleosides by incubation in a mixture of benzonase (Santa Cruz Biotechnology, sc-391121B), nuclease P1 from *Penicillium citrinum* (Sigma, N8630), and alkaline phosphatase from *E. coli* (Sigma-Aldrich, P5931) in 10 mM ammonium acetate, pH 6.0, 1 mM magnesium chloride buffer at 40 °C for 40 min. Three volume equivalents of ice-cold acetonitrile were added to the reactions after digestion was completed to precipitate proteineous contaminants. Following centrifugation at $16000 \times g$ at 4 °C for 40 min, the supernatants were collected in new tubes and dried under vacuum at room temperature. The resulting residues were dissolved in water for HPLC-MS/ MS. Chromatographic separation was performed using a Shimadzu Prominence LC-20AD HPLC system with an Ascentis Express C18 2.7 µm 150 ×2.1 mm i.d. column equipped with an Ascentis Express Cartridge Guard Column (Supelco Analytical, Bellefonte, PA, USA) with EXP Titanium Hybrid Ferrule (Optimize Technologies Inc.). For analysis of unmodified nucleosides the following conditions were applied: isocratic flow consisting of 75% A (0.1 % formic acid in water) and 25% B (0.1 % formic acid in methanol) at 0.16 ml/min, 40 °C. For analysis of 5-ohC: 0.14 ml/min flow starting with 5% B for 0.5 min, followed with a gradient of 5–45% B for 7.5 min, finishing with re-equilibration with 5% B for 5.5 min. Online mass spectrometry detection was performed using an Applied Biosystems/MDS Sciex API5000 Triple quadrupole mass spectrometer (ABsciex, Toronto, Canada), operating in positive electrospray ionization mode. The deoxyribonucleosides were monitored by multiple reaction monitoring using the following mass transitions (m/z): $252.1 \rightarrow 136.1$ (dA), $228.1 \rightarrow 112.1$ (dC), $268.1 \rightarrow 152.1$ (dG), $243.1 \rightarrow 127.1$ (dT), and $244.1 \rightarrow 128.1$ (5-ohdC).

*Single-cell gel electrophoresis (SCGE) / alkaline comet assay.* A modified SCGE / alkaline comet assay was performed as previously described[72] in a high-throughput format[73]. Six-month-old male mice were sacrificed and the left hippocampus rapidly dissected using a stereomicroscope. The tissue was immediately placed in ice-cold isotonic solution (Merchant's buffer: 0.14 M NaCl, 1.47 mM $KH_2PO_4$, 2.7 mM KCl, 8.1 mM $Na_2HPO_4$, 10 mM NaEDTA, pH 7.4, containing EDTA to inhibit cleavage of DNA), mechanically minced to obtain a single-cell/nuclei suspension and filtered (100 µm nylon mesh)[74,75]. The single-cell suspensions were counted (Invitrogen Countess™) and diluted to densities appropriate for SCGE ($1 \times 10^6$ cells/ml). Cell suspensions were mixed 1:10 with 0.75% Low Melting Point agarose (Gibco BRL 5517US) in PBS, pH 7.4, w/o calcium and magnesium, with 10 mM $Na_2EDTA$, to a final agarose concentration of 0.67%. Aliquots of the cell/ agarose mixture were instantly added to cold polyester films (GelBond®). Solidified gels were immediately immersed in lysis solution (2.5 M NaCl, 0.1 mM EDTA, 10 mM Tris, 1% Sodium Lauryl Sarcosinate, with 1 ml Triton X-100 and 10 ml DMSO per 100 ml solution). After lysis at 4 °C overnight, films were washed 1 × 10 min and 1 × 50 min in cold enzyme buffer (40 mM HEPES, 0,1 M KCl, 0,5 mM EDTA, pH 8.0) at 4 °C prior to enzyme treatment. To detect oxidative DNA base lesions, we used the well-characterized *E. coli* DNA repair enzyme, Formamidopyrimidine DNA glycosylase (Fpg), as previously described[72,76–82]. Fpg (1 µg/ml) and BSA (0.2 mg/ml) were added to prewarmed enzyme buffer, in which films were immersed and incubated for 1 h at 37 °C. Control films were treated similarly, but with enzyme buffer only (no Fpg added). The Fpg-concentration was optimized based on titration experiments with a photoactivated drug (Ro12-9786) plus cold visible light. After enzyme incubation, films were immersed in cold electrophoresis solution (0.3 M NaOH, 0.1 M EDTA, > pH 13.2) for 5 min + 35 min for unwinding, and electrophoresis was carried out for 25 min at 8–10 °C. The voltage potential was 0.80–0.90 V/cm across the stage. Subsequently, films were neutralized in Tris-buffer (0.4 M Tris, pH to 7.5) for 2 × 5 min, rinsed in $dH_2O$, fixed in 96% EtOH for 1.5 h, and dried overnight. Films were rehydrated for 20 min at room temperature in TE-buffer pH 7.5, containing 10,000 × diluted SYBRGold stain (Molecular Probes), under gentle shaking. The films were rinsed in $dH_2O$ and covered with large coverslips (80 × 120 mm, thickness no.1, VWR International AS, Oslo, Norway). Imaging was performed with an epi-fluorescence microscope (Olympus BX51). Semiautomated scoring of 50 comet tails per gel was done with "Comet assay IV" software (Perceptive Instruments Ltd, UK). A cell exhibiting 0% tail intensity has no DNA in the tail, and hence no detectable DNA damage under the conditions used. Increasing fluorescence in the tail vs the head of the comet indicates increasing DNA damage levels, up to a level of 100%, where the entire DNA is present in the tail. The median comet tail intensity per sample (50 comets × 3 replicate gels scored) was used to calculate the mean values per genotype. Net Fpg-sensitive sites were calculated by subtracting the median comet tail intensity for samples incubated without Fpg from those treated with Fpg.

*PCR-based DNA damage detection.* Hippocampal DNA damage levels were quantified by using a restriction enzyme-based qPCR method[83]. Briefly, DNA damage in a TaqI-sensitive restriction site will result in altered cutting frequency of the DNA, which ultimately will affect PCR amplification of a target sequence spanning the restriction site. Total genomic DNA was isolated from the hippocampus of 6-month-old male mice using the DNeasy Blood and Tissue kit according to manufacturer's protocol (Qiagen, cat. no. 80004). 30 ng of DNA was subjected to $Taq^\alpha$I restriction enzyme digestion followed by qPCR amplification of a target sequence in the *Gapdh* gene. The *Gapdh* gene is used because it is transcriptionally active and thus, easily accessible for damage induction, but also repair. Relative amounts of PCR products, reflecting the level of damage in each sample, were calculated by using the comparative ΔCT method. Primers: *Gapdh* forward, 5′ cttcaacagcaactcccact and reverse, 5′aaaagtcaggtttcccatcc.

## DNA mutation analysis

*Whole-genome deep sequencing.* For each genotype, hippocampal genomic DNA from four 6-month-old male mice was isolated using DNeasy Blood and Tissue Kit (Qiagen, cat. no. 80004), pooled and sent to BGI Tech Solutions, Hong Kong, for whole-genome sequencing, including library construction and HighSeq4000 sequencing.

*Identification of strain-dependent genetic variations.* We identified SNPs and insertions/deletions (InDels) individually for mutant and WT samples. Specifically, the adapter sequence in the raw data was removed, and low-quality reads which had too many Ns (>10%) or low-quality score (<5) was discarded. The remaining reads were aligned to the mouse reference sequence (mm10) using the Burrows-Wheeler Aligner (BWA)[84]. The alignment information was stored in BAM format files, which was further processed by fixing mate-pair information, adding read group information and marking duplicate reads caused by polymerase chain reaction artefacts. The variant calling steps included SNPs detected by SOAPsnp[85] and small InDels detected by Samtools/GATK[86]. In GATK, the caller *UnifiedGenotyper* was used with the parameters *stand_call_conf* set to 50 and *stand_emit_conf* set to 10. Hard filtering was applied to get variant results of higher confidence. To identify strain-dependent genetic variation—i.e., variants inherited from the 129 strain and not completely lost through back-crossing with the C57BL/ 6 N strain—SNP and InDel data were loaded into the genome browser *SeqMonk* (http://www.bioinformatics.babraham.ac.uk/projects/seqmonk/) for further inspection. We defined 129-specific regions as having more than 50 detected SNPs or InDels per 600 kB bases and used this as a criterion in the "Read Count Quantitation using all Reads" probe extraction method in *SeqMonk*. Individual regions satisfying this criterion were extracted and consecutive regions within the genome were joined to form the final 129-dependent regions. We confirmed enrichment of 129-dependent genetic variants within each region by identifying the SNPs that were present in dbSNP (build 137) and counting the number of times the SNP genotype matched the annotated 129 (129P2/OlaHsd, 129S1/SvImJ, or 129S5SvEv strains) or black 6 (C57BL/6NJ strain) genotypes.

*Identification of mutations in NEIL-deficient hippocampi.* Reads were filtered and aligned to the mouse genome as described above, and alignments were pre-processed according to GATK Best Practices recommendations[87] using GATK version 3.5, including local realignment around InDels and recalibration of quality scores. For calling we used the MUTECT2 variant caller[88], with KOs as case and WT as control. Briefly, MUTECT2 identifies variants that are present in the case sample but are absent in the control sample and where the difference is unlikely due to sequencing errors. We used MUTECT2 default parameters, which include rejecting candidates that in the control sample have (i) supporting reads numbering ≥ 2 or constituting ≥ 3% of the total reads (i.e., <34 total reads) and (ii) their quality scores sums to >20. We used snpEff[89] and SnpSift[90] to annotate all SNPs and InDels found and discarded SNPs and InDels overlapping the 129-specific intervals for each sample.

## Electrophysiology

*Slice/Sample preparation.* Adult (4-month-old) WT and $Neil1^{-/-}Neil2^{-/-}$ male mice were sacrificed with Suprane (Baxter) and the brains removed. Transverse slices (400 µm) were cut from the middle and dorsal portion of each hippocampus with a vibroslicer (Leica VT 1200) in artificial cerebrospinal fluid (ACSF, 4 °C, bubbled with 95% $O_2$–5% $CO_2$) containing (in mM): 124 NaCl, 2 KCl, 1.25 $KH_2PO_4$, 2 $MgSO_4$, 1 $CaCl_2$, 26 $NaHCO_3$ and 12 glucose. Slices were placed in an interface chamber exposed to humidified gas at 28–32 °C and perfused with ACSF (pH 7.3) containing 2 mM $CaCl_2$ for at least 1 h prior to the experiments. In some of the experiments, DL-2-amino-5-phosphopentanoic acid (AP5, 50uM; Sigma-Aldrich, Oslo, Norway) was added to the ACSF in order to block NMDA-receptor-mediated synaptic plasticity.

*Synaptic transmission, synaptic excitability and paired-pulse facilitation.* Orthodromic synaptic stimuli (<300 µA, 0.1 Hz) were delivered through tungsten electrodes (0.1 MOhm WPI, USA) placed in the middle of either SR or SO of the hippocampal CA1 region. The presynaptic volley and the field excitatory postsynaptic potential (fEPSP) were recorded by a glass electrode (filled with ACSF)

placed in the corresponding synaptic layer (separated approximately 200 μm from the stimulation electrode) while another electrode placed in the pyramidal cell body layer (SP) monitored the population spike. Following a period of at least 20 min with stable responses, we stimulated the afferent fibers with increasing strength (increasing the stimulus duration in steps of 10 μs from 0 to 90 μs, five consecutive stimulations at each step). Prior to the input/output (I/O), the strength was adjusted so that a population spike appeared in response to 40, 50, or 60 μs in order to define the stimulation/response range. A similar approach was used to elicit paired-pulse responses (50 ms interstimulus interval, the two stimuli being equal in strength). To assess synaptic transmission, we measured the amplitudes of the presynaptic volley and the fEPSP at the different stimulation strengths. During the analysis, care was taken to use extrapolated measurements, which were within the apparently linear part of the I/O curves. Values from individual experiments outside the linear part of the I/O curves (prevolley vs. fEPSP) were omitted when pooling the data. The population spike amplitude was measured as the distance between the maximal population spike amplitude and a line joining the maximum pre- and postspike fEPSP positivities. In order to pool data from the paired-pulse experiments, we selected responses to stimulation strength just below the threshold for eliciting a population spike on the second fEPSP.

*Long-term potentiation (LTP) of synaptic transmission.* Orthodromic synaptic stimuli (50 μs, <300 μA) were delivered alternately through two tungsten electrodes (0.1 MOhm WPI, USA), one situated in the middle of SR and another in the middle of SO of the hippocampal CA1 region close to the CA3 border. Extracellular synaptic responses were monitored by two glass electrodes spaced approximately 200 μm from the stimulation electrode (2–5 MOhm filled with ACSF) placed in the corresponding synaptic layers. After obtaining a stable synaptic response in both pathways (0.1 Hz stimulation) for at least 10–15 min, one of the pathways was tetanized (a single 100 Hz tetanization for 1 s, repeated four times at 5 min intervals). As standardization, the stimulation strength used for tetanization was just above the threshold for generation of a population spike in response to a single test shock. Synaptic efficacy was assessed by measuring the slope of the fEPSP in the middle third of its rising phase. Six consecutive responses (1 min) were averaged and normalized to the mean value recorded 1–4 min prior to tetanization. Data were pooled across animals of the same experimental group and pathway and are presented as mean ± SEM. All experiments were done in parallel in two separate electrophysiology setups. One setup was equipped with two Axoclamp2B amplifiers (low pass filtered at 3 kHz) (Molecular Devices, USA), custom-made 10X amplifiers (in house), Digitizer (National Instruments, USA) and custom-made programs for recording and analysis. The other setup was equipped with one Axoclamp2A amplifier (Molecular Devices, USA) and one Bio-logic VF180 (Claix, France) (Both low pass filtered at 3 kHz), custom-made 10X amplifiers (in house), Digitizer (National Instruments, USA), and custom-made programs for recording and analysis.

### Laser capture microdissection and RNA sequence analysis

*Tissue processing.* Mouse brains were isolated from 3–6-month-old male mice without prior perfusion within 150 +/−30 s, mounted onto cryostat sockets (Leica CM3050S, Nussloch, Germany) and frozen in liquid nitrogen immediately. Subsequently, samples were stored in liquid nitrogen until further processing (max. 1 h). Each brain was completely processed the same day according to the workflow described below. Brains were cut in coronal orientation at a thickness of 8 μm using a cryostat (Leica CM3050S, Nussloch, Germany), starting from the onset of the hippocampal formation until the end. The Allen Mouse Brain Atlas (Allen Brain Institute) was used as a refs. [20–30]. laser dissection membrane slides (Molecular Machines and Industries GmbH, Eching, Germany) with 5–7 brain slices each were collected from each brain. All slides were subsequently used for tissue collection to avoid a collection bias alongside the rostro-caudal axis.

*Tissue collection.* CA1 dissectates were collected using a laser dissection microscope (Molecular Machines and Industries, CellCut on Olympus IX71, Eching, Germany). Only dorsal hippocampus was included in the tissue collection described in the following. The hippocampal CA1 area was identified using the Allen Mouse Brain Atlas as a reference. For each slice, the whole CA1 area was defined manually (see Fig. 4a). We collected 20 CA1 dissectates in one isolation cap (Molecular Machines and Industries GmbH, Eching, Germany) before adding RLT lysis buffer (AllPrep Kit, Qiagen, Hilden, Germany). Samples from one individual were collected the same day. Samples were vortexed shortly and frozen at −80 °C until further processing.

*RNA extraction.* RNA was isolated using the RNeasy Mini Kit (Qiagen, Hilden, Germany) according to the manufacturer's instructions. Samples were briefly vortexed before starting the RNA isolation steps; no additional tissue lysis procedure was performed. RNA samples yielded >280 ng of RNA ( > 5.6 ng/μl in a total eluate of 50 μl) with a RIN value of generally > 7 as determined by Bioanalyzer (Agilent Technologies, Santa Clara, USA).

*RNAseq.* Whole-transcriptome sequencing was done by BGI Group (BGI Genomics Co., Ltd., Hong Kong, China). In brief, the steps were as follows: a) quality

control using the Agilent 2100 Bio analyzer (Agilent RNA 6000 Nano Kit, Agilent Technologies, Santa Clara, CA, USA), b) purification of poly-A containing mRNA by poly-T oligo-attached magnetic beads, c) mRNA-fragmentation (divalent cations, high temperature), d) reverse transcription into cDNA, e) Qubit quantification (Thermo Fisher Scientific, Waltham, MA, USA), f) library construction by using a single strand DNA circle (see library construction quality control results, Supplementary Fig. 3h), g) rolling circle replication creating DNA nanoballs (more fluorescent signal during sequencing), h) reading of pair-end reads of 100 bp via BGISEQ-500 platform[91].

*RNAseq bioinformatic preprocessing.* Bioinformatic processing was in part done by BGI (BGI Genomics Co., Ltd., Hong Kong, China) using the following workflow: a) filtering of low-quality reads using the software SOAPnuke[92], b) genome mapping with HISAT software[93], c) transcript reconstruction using StringTie[94] and reference comparison with Cuffcompare[95], d) prediction of coding potential with CPC[96], e) SNP and INDEL detection with GATK[86], e) reference mapping with Bowtie2[97], f) calculation of gene expression levels using the RSEM software package[98], g) hierarchical clustering with hclust in R.

*Analysis of differential expression.* We created the count matrix of integer values and the metadata matrix based on the sequencing results from BGI (un-normalized estimated counts). We next performed a DESeq2 differential gene expression analysis (R version "Dark and Stormy Night" for Windows[99] comparing between the different genotype groups (n = 3 mice per group; *Neil1*−/−, *Neil2*−/−, *Neil1*−/−*Neil2*−/−, WT). Low counts were filtered using rowSums function. The alpha parameter of the results function was set to 0.05 and adjusted *p* value was calculated using Benjamini-Hochberg correction for multiple testing (see all DEGs Supplementary Fig. 3d–f). We performed this analysis after prior exploratory data analysis based on the BGI-results, which showed two samples to be clear outliers from the remaining replicates (PCA plot, Supplementary Fig. 3g). These two outliers (red arrows) were excluded from the group comparison analysis (Supplementary Fig. 4, PCA plot without outliers). For the over-representation analysis the online version of PANTHER Classification System release 15.0 was used[100]. We chose a Binomial/FDR multiple testing correction, both for the gene ontology biological processes term enrichment analysis and the reactome-pathway enrichment analysis (Reactom version 73 2020/06/11)[101]. All DEGs log2fold > 0.3 (fold > 1.23) were included in the enrichment analysis (Supplementary Fig. 3d–f).

### Immunohistochemistry

*Mouse perfusion.* Male mice (3–6-month-old) were anesthetized and sacrificed using first isoflurane (Baxter, Oslo, Norway) and subsequently an intraperitoneal, weight-adapted overdose of pentobarbital (>200 mg/kg body weight). Intracardial perfusion was done with 0.9% saline (B.Braun, Melsungen, Germany) and 4% paraformaldehyde in phosphate-buffered saline (PBS). Brains were put into 4% paraformaldehyde/PBS solution for a minimum of 48 h at 4 °C for fixation. We sectioned brains at a thickness of 30 μm using a cryostat (Leica CM3050S, Nussloch, Germany) starting at a medio-lateral depth of 900 μm and continuing until the end of the tissue block. Slices were then stored at 4 °C in a PBS-solution containing 0.05% of Proclin (Merck, Darmstadt, Germany) until further processing.

*Antibody treatment.* Heat-mediated antigen retrieval was performed for 3 min at 99 °C in a 40 mM trisodium citrate (Merck, Darmstadt, Germany) solution, pH 6.0. Specimens were then left to cool down to room temperature inside this solution for another 27 min (30 min total exposure). 5% normal goat serum/bovine serum albumin served as a blocking agent against unspecific binding. We then immediately incubated in primary antibody solution (GABRA2 rabbit polyclonal, 1:1000 dilution, Cat.No.224103, Synaptic Systems, Goettingen, Germany; NR2A rabbit polyclonal, 1:250 dilution, Cat.No. AGC-002, alomone labs, Jerusalem, Israel; NR2B rabbit polyclonal, 1:100 dilution, Cat.No. AGC-003, alomone labs, Jerusalem, Israel; NeuN mouse monoclonal, A40/MAB377, 1:500 dilution, Cat.No. 636574, ThermoFisher, Waltham, MA, USA; PSD-95 rabbit polyclonal, 1:500 dilution, Cat.No. PA585769, ThermoFisher, Waltham, MA, USA) overnight at 4 °C (exception: PSD-95 2 h at room temperature, no antigen retrieval step) under constant shaking at 15 oscillations/min. We generally used Alexa Flour dyes (A488, A555, A647; 1:1000 dilution) as secondary antibodies (Thermo Fisher Scientific, Waltham, MA, USA) and DAPI (Merck, Darmstadt, Germany; 1:5000 dilution) as a non-specific nuclear counter stain.

*Confocal microscopy.* Imaging of stained slices was done using a Zeiss LSM880 confocal microscope. For synaptic markers, a Plan-Apochromat 40x/1.4 Oil DIC M27 objective (Carl Zeiss, Jena, Germany) was used. An imaging square of 700 × 700 μm (x/y 2000pixels of 0.35 μm each) and a z-interval of 0.5 μm was applied. A proximal and a distal imaging square was set within the CA1 region based on NeuN and DAPI stainings as an anatomical orientation (center of proximal square set at 0.25x total CA1 length measured from proximal end, center of distal square at 0.75xtotal CA1 length). Results displayed are averaged across proximal and distal squares. For each animal, one medial and one lateral brain slice were analyzed, amounting to a total of 4 imaging squares (2 proximal, 2 distal).

With respect to the nested data problem[102], all statistical analysis was done at an animal level (1 animal = 1 statistical unit). On a sideline, we observed a consistently different NeuN-signal in NEIL2-deficient mice. Tissue quality, immunostaining protocol parameters, background signal and DAPI-counter-staining efficiency was identical in these samples compared to the other genotypes, so that a genotype-specific NeuN-signal appears possible and, while beyond the scope of this manuscript, warrants further investigation.

*Quantification of immunoreactivity*. Imaris 9.3 (Bitplane, Zurich, Switzerland) was used to quantify immunoreactivity. We first created a 3D reconstruction of the whole z-plane dataset. The strata pyramidale/oriens/radiatum were identified as regions of interest using the "surface" tool in Imaris and copied to every z-plane accordingly. Based on the surface selection, a 3D-frame was created and the parameter of interest "masked" according to this frame. The Imaris "Spots Wizard" function was used to identify areas of synaptic reactivity within this masked channel (1 μm spot diameter, background subtraction applied according to local contrast). We then conducted a pilot-experiment for every immunohistochemical marker used, involving typically 4 different images. This was to make sure a biologically relevant signal is captured by the software. Based on this pilot experiment, a selection criterion based on the "quality" (see bitplane.com/imaris) filter in Imaris was defined and kept the same throughout the analysis. Imaris automated background subtraction was done for every specimen analyzed to account for intensity variations despite identical immunohistochemistry and confocal parameters. 2/3 of the region of interest had to be intact (i.e. not damaged by tissue cracks, covered by imaging artefacts etc.) in order to be included in the analysis. One exclusion was made based on this criterion (NR2A immunostaining, wildtype animal). No outlier correction was performed before statistical analysis. No image processing was performed prior to Imaris quantification (raw image data, Zeiss "lsm" file format). Supplementary Fig. 6 shows a background-filtered signal indicating the approximate portion detected by the spot-detection tool.

**Statistics and reproducibility**. In Figs. 1b–i, 2b, Supplementary Figs. 1 and 2, statistical evaluation was done by one-way ANOVA followed by Tukey's multiple comparisons test using GraphPad Prism version 6.07 for Windows (GraphPad Software, La Jolla California USA, www.graphpad.com). Data in Fig. 1j, k–l were analyzed by non-parametric pairwise Wilcoxon rank-sum test, Holm-adjusted, and post-hoc family-wise multiple comparison of means (Tukey Honest Significant Difference), respectively, using R version 3.0.2. In Fig. 2c, the data were analyzed by two-way ANOVA, followed by Sidak's multiple comparisons test using GraphPad Prism version 8.4. In Fig. 2d–g the MuTect2 (Genome Analysis Toolkit (GATK) toolset) software was used. In Fig. 3, statistical evaluation was done by linear mixed model analysis, using SAS 9.2 (SAS Institute Inc.). For statistical analysis of RNA seq data (Fig. 4 and Supplementary Fig. 3), we refer to the Methods section. Data in Supplementary Figs. 6a, b and 7a were analyzed by two-way ANOVA and in Supplementary Fig. 7b by one-way ANOVA, followed by Tukey's multiple comparisons, using GraphPad Prism version 8.4. In Supplementary Fig. 6c, multiple t-tests with Holm-Sidak correction, was applied. A p-value < 0.05 was considered significant. The number of replicates and how they are defined are given in respective figure legends.

Regarding statistical power considerations, we based our calculations on a difference in means (immunohistochemistry-experiments) of ca 50% and an SD of about 20%. These approximate assumptions are based on differences in NMDA-receptor subunit reactivity observed in experiments conducted in NEIL3-deficient animals[14]. This resulted in a sample size of minimally 3 per group, ideally 6 per group, depending on the assumed SD. In the experiments presented in this study, we chose a 3 × 2 approach, i.e. 2 brain slices (1 medial, 1 lateral) per animal in a total of three animals per group, which proved to be an efficient strategy in avoiding outliers and accounting for staining differences in the mediolateral brain axis. We stringently paid attention to the nested data problem[102] and all the immunohistochemistry analyses are based on the assumption that 1 animal is 1 statistical unit. As stated before, we used the DESeq2-tool[99] to analyze the transcriptome data, which statistically accounts for relatively small sample sizes of n = 3, which are typical in technically challenging experiments as the one presented in this study. As for the behavior and electrophysiology experiments, a priori power calculations proved to be difficult due to limited statistical data in similar transgenic mouse models. However, a post-hoc power calculation using the tool clincalc.com yielded approximate power-values of 80–90% based on n = 15–20 and an alpha-value of 0.05 (electrophysiology) and 90–100% based on n = 10–43/alpha 0.05 (behavior). Approximate power-ranges are between the group with the highest "n"/lowest variance and the group with the lowest "n"/highest variance. As is the rule in electrophysiology experiments of this type, several brain slices "n" were harvested per individual mouse.

**Reporting summary**. Further information on research design is available in the Nature Research Reporting Summary linked to this article.

## Data availability

The RNA sequence data generated and analyzed during the current study have been deposited in NCBI's Gene Expression Ominbus[103], accession number GSE160621. The DNA sequence data have been deposited in European Nucleotide Archive (ENA), accession number PRJEB31108. The figure source data are available in Supplementary Data 1. Further information and requests for resources and reagents should be directed to and will be fulfilled by the Lead Contact, Magnar Bjørås (magnar.bjoras@ntnu.no).

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

## Acknowledgements

We thank our colleague and friend Øivind Hvalby† who actively contributed to the electrophysiological data collection and analysis in this study but sadly passed away in 2015. We thank Atle van Beelen Granlund for his help in setting up the laser capture microdissection experiment. The LC-MS/MS analysis of DNA modifications was performed by the Proteomics and Modomics Experimental Core Facility (PROMEC), Norwegian University of Science and Technology (NTNU). This work was sponsored by the Research Council of Norway [287911, 240529]; the South-Eastern Norway Regional Health Authority [46060921, 2017117]; and partly supported by the Research Council of Norway through its Centers of Excellence funding scheme [223268/F50]. PROMEC is funded by the Faculty of Medicine and Health Sciences at NTNU and the Central Norway Regional Health Authority.

## Author contributions

M.B., K.S., J.Y., A.K., L.E., and G.S. directed the study and/or obtained financial support. G.A.H. and L.L. generated the NEIL1 KO mouse model in collaboration with the Norwegian Transgenic Center. R.S. generated the NEIL2 KO mouse model and backcrossed and bred the mice. N.K. backcrossed and bred the mice in Trondheim. V.R., O.M. and M.D.S. designed behavioral experiments. V.M., R.S., S.V. and M.D.B. conducted behavioral experiments. A.D.R. analyzed behavioral results, including statistics. An.K. performed HPLC-MS/MS. A.K.O. and K.B.G. performed alkaline comet assay. K.S. performed PCR-based DNA damage assay. P.S. and K.S. analyzed DNA sequencing data. V.J. designed and conducted electrophysiological studies and analyzed the data obtained. N.K. designed the laser capture microdissection experiment. N.K., S.B.S. and M.S.F. performed the laser capture microdissection experiment. N.K. isolated RNA from laser-dissected samples. M.S.F. performed the immunostaining experiments. N.K. did the confocal imaging and subsequent Imaris-analysis as well as statistical analysis. W.W. performed qPCR. A.M.B. analyzed the RNAseq data acquired from BGI. A.M.B. and N.K. performed the gene ontology and pathway analysis of RNAseq data. N.K. did the single-gene QuickGo thematic analysis. N.K. and G.A.H. layouted, revised and structured all figures based on single plots and figures made by other authors. G.A.H., V.R., N.K, K.S. and M.B. wrote the manuscript with input from the other authors.

## Competing interests

The authors declare no competing interests.
