## [Transparent Peer Review File · Communications Biology]

Reviewers' comments:

Reviewer #1 (Remarks to the Author):

The study of Hildrestrand et al., follows a number of studies that document phenotypes of mice deficient in DNA glycosylases operating in base excision repair. The mice have no/mild phenotypes, presumably because of functional overlap of different glycosylases. The main novelty of this paper, is the analysis of Neil1 Neil2 double knockout mice. The data from Neil1-KO mice are consistent with previous findings: increased weight starting in adulthood reflecting a mild metabolic syndrome (Vartanian, PNAS 2005); increased mobility in an open field test. The weight gain phenotype did not occur in Neil2-KO and double mutants, while double mutants showed a further increase in spontaneous mobility in open field as well as in an elevated maze. In addition, this study documents subtle changes in Morris water maze tests, hippocampal slice electrophysiology and hippocampal CA1 transcriptome, and no change in oxidative DNA damage markers. The data are relevant in that they reproduce previous findings, showing that something is going on in Neil1-KO mice and that this phenotype is modified by the absence of Neil2, and is not obviously linked to the accumulation of DNA damage. A major problem is that data are overinterpreted.

Specific comments:

1) Title, abstract and rest of the paper: Overall the authors have a tendency to oversell the data; I would strongly recommend to tone down the conclusions and present the data as they are. Statements like 'NEIL1 promotes spatial memory retention ...' (Canugovi et al., PNAS 2012) are misleading and disorienting young scientist. What is shown in that paper is that Neil1-KO mice display subtle changes in Morris water maze performance that are difficult to interpret and that by themselves are insufficient to draw any conclusions about a role of NEIL1 in spatial memory retention. Nevertheless, based on that tittle, NEIL1 is sold as a learning and memory factor in the literature, and perhaps also has distracted some of the scientists contributing to this paper. Pleas tone down the title, abstract, and rest of the paper!! There is no data about learning and anxiety in the manuscript, nor is there evidence for a direct role of NEIL1 nor NEIL2 in hippocampal neurons. As a starter, statements about altered 'anxiety' in the open field require changes in the time spent in the center area, (eventually complemented by other parameters such as fecal boli deposits, etc). However, time in the center area is unaltered in mutants (Fig. 1H). Thus, basically the data indicate that, if anything, the mutants show unaltered anxiety status. Likewise, the MWM data do not show changes in key parameters for learning and memory consistent with unaltered LTP. The data suggest altered time spend in specific non-target quadrants. These data are difficult to interpret, and can not be linked to memory performance. Changes in behavior in KO mice (in particular global KO mice) do not necessarily uncover the function of the eliminated protein. Since elimination of NEIL1 alters the metabolic and perhaps the health status of the animals, this may influence behavioral readouts. In any case, the data do not support statements such as 'NEIL1 and NEIL2 regulate ...' (tittle) and 'NEIL1 and NEIL3 alter cognition ..' (abstract). Please, be more precise: 'the absence of NEIL1 and NEIL2 affectsbehavior, gene expression etc....' Also be more explicit about readouts that do not changes, rather than overselling marginal changes in non-relevant and poorly documented changes such as 'reduced stratum oriens afferents...'. .

2) Figure 1 and 2

For behavioral experiments provide more relevant details in methods/results section. I have the impression that single and double mouse lines are bred independently and that 'controls' are not littermates, generated from mating of heterozygotes, which would be methodologically more correct. Are mice group housed during the experiments; are the same mice used for different experiments, and if so, what is the order of experiments. Is the same cohort used for DNA damage analysis?

3) Figure 4 and S4

RNAseq in combination with laser dissection is more noisy than other approaches, and the experiments seemingly was poorly designed resulting in insufficient samples of Neil1 and double KO mice. This may explain more significant changes in Neil2-KO. Is there a PC-plot without the 'outliers'? In PC-plot in Fig. S4G use different symbols (in addition to colors) to indicate different mouse types. Tone down title of this section: the DEG have a very putative, indirect link with synaptic function: just state that Neil1 and Neil2 KO have a differential effect on CA1 gene expression

4) Figure 5 and S5

These data are fully inconclusive: Synapses of CA1 pyramidal neurons localize in the stratum radiatum, moleculare and oriens rather than in the soma of the pyramidal neurons. What is stained here is not synaptic proteins. Please use Western blot to demonstrate changes in NMDA and GABA receptor status There are also problems with the reasoning. If somehow Neil2 via Nr1d1 etc. regulates synapses this also would result in changes in receptor mRNA's.

5) The data on effects of Neil KO on weight are not discussed. What happens in double KO compared to Neil1 KO?

Reviewer #2 (Remarks to the Author):

NEIL1 and NEIL2 are enzymes which display glycosylase activity and take part in the initial steps of the base excision repair (BER) pathway. Several studies suggest that BER plays an important role in protecting the brain from oxidative DNA damage. In this manuscript by Hildestrand et al., the authors investigate a possible non-canonical function of NEIL1 and NEIL2 in modulating memory formation and anxiety in mice. Based on their observations they postulate that within CA1 of the hippocampus NEIL1 and NEIL2 jointly regulate transcription of genes relevant for synaptic function. Overall, this is an interesting report which sheds light on prospective alternative roles of the NEIL1/2 glycosylases in brain in addition to their roles in the canonical BER pathway. The mechanism behind this function is, however, not fully elucidated.

Overall the manuscript is well written and the experimental work is appropriately described. Nevertheless, some issues do need consideration as described below.

Comments:

Some of the results obtained in this study, including accumulation of DNA damage, may depend strongly on the age of the mice. Therefore, it is important that the age of the mice in this study is clearly stated in the Results section and that potential issues regarding age are discussed in more depth in the Discussion.

The authors should discuss in more detail the difference between what is measured in the open field test and the elevated zero maze. This is an important issue since significant differences between WT and NEIL deficient mice are observed only in Fig 1I but not in Fig 1H. Accordingly, the authors should modify their statement in Line 278-280, where they claim that their study reveals reduced anxiety in the open field test and the elevated zero maze.

In line 107-109 it says: during the 2nd retention trial (day 12), the NEIL1-/-NEIL2-/- mice became less decisive, shown as decreased preference for the target zone (Fig 1K). The authors should clearly state whether this is just a trend or statistically significant and the observed differences between day 4, 5 and 12 should be discussed in more detail.

The authors suggest that the observed changed behaviour in the KO mice may be due to a role for NEIL1/NEIL2 in epigenetic gene regulation. In order to test this hypothesis, the authors should

consider to compare the methylation pattern for the *Npas4* and *Nr4a1-3* gene loci (or other DEGs), respectively, in the WT and the KO mice.

Comments for specific figures:

Figure 1 A: The authors should consider to revise the illustration of the elevated zero maze – the perspective of the current figure does not seem to be correct

Figure 1 J-K: The figure legend lacks information regarding colour code (shades of grey and black) for the round symbols.

Figure 2B: The authors should comment on why they have chosen to focus on 5-ohC and not e.g. 5-ohU when estimating damage accumulation. Is one type of adduct more common or a better substrate for NEIL1 or NEIL2 than the other, or?

Figure 2 C: From reading the Materials and Methods section and the figure legend it is not clear what would be equal to 100% comet tail intensity. Also, it does not seem obvious, why the tail intensity appears to be lower and not higher after treatment with Fpg. Please clarify.

Figure 2 D: The 600 SNPs, which are present in the double KO, but not in the single KO mice – are these evenly distributed in different regions of the genome?

Figure 2 E: Previous studies of NEIL1 and NEIL2 point to a potential role for these enzymes in transcription associated DNA repair. Therefore, it would be interesting to also present the data from the current study in a way, where the genes are separated into expressed and non-expressed genes, respectively, for each of the genotypes.

Figure 2 G: Indication of colour code (blue, green and purple) is missing

Figure 4 B: It looks as if the difference in e.g. total # DEGs between NEIL1 *-/-* and NEIL2 *-/-* is significant – but why does this diagram does contain any indications of significance?

Minor points:

In several cases periods are lacking at the end of sentences (after a reference index number).

The layout for the references is not the same throughout the manuscript

Abbreviations are not always introduced or explained first time they are used

Line 36: the authors should consider to write “..., while the role of NEIL2 remains unclear”

Line 77-78: the authors should consider to write a few more words to explain what is meant by “mild metabolic phenotype”

Line 95-96: the authors should consider to write “by being more mobile” instead of “by being more active”, since they initiate the sentence by writing “mice displayed hyperactive behaviour”

Line 112: the authors should also comment on the observed significant differences between the two single ko mouse genotypes and between the NEIL1 *-/-* and NEIL1-NEIL2 double ko mice shown in Figure S1

Line 133-136: it would be appropriate to emphasize that for the data presented in Figure S2 the level of damage is measured in a single specific gene and not in the whole genome

Line 136: "hippocampus" should be "hippocampi"

Line 138: Figure S4 is mentioned before Figure S3 (Line 195)

Line 140: "suggest" should be "suggests"

Line 152 and 154: the text refers to Figure 4E and 4F. Do they mean 2E and 2F?

Line 167 and 170: the authors should consider to write "intensities" instead of "strengths"

Line 251: abbreviation instead of full-length word for "stratum pyramidale" should be used as the abbreviation has been introduced earlier

Line 253, 359 and 362: abbreviation instead of full-length word for "stratum oriens" should be used as the abbreviation has been introduced earlier

Line 281: in this line it is not completely clear whether the authors refer the NEIL1/2 deficient mice or the OGG1/MUTYH mice, therefore this sentence should be edited for clarification.

Line 317-318: the authors should elaborate on their sentence "we may speculate that this is caused by a putative role of NEIL2 in processes making the chromatin more accessible to strand breaks"

Line 451: "using following" should be "using the following"

Line 668: the authors should consider to use the word "sacrificed" instead of "killed"

Reviewer #3 (Remarks to the Author):

DNA glycosylases play important roles in the repair of DNA resultant from oxidative damage and thus participate in the prevention of neuronal death after ischemia and age-related cognitive decline. The present manuscript describes the effects of deletion of two DNA glycosylases (NEIL1 and NEIL2) on the behavior of single and double knockout (KO) mice and evaluated potential underlying mechanisms.

Previous work by this group have provided evidence that deletion of the DNA glycosylase NEIL3 resulted in impaired learning and memory and reduced anxiety levels which was attributed to reduced proliferative capacity of neural stem cells due to impaired DNA repair capacity. They now report that mice lacking both NEIL1 and NEIL2 glycosylases do not show impaired DNA oxidative repair mechanism but exhibited reduced anxiety and improved learning and memory compared to control wild-type (WT) mice. These behavioral alterations were attribute to reduced hippocampal stratum oriens afferents into pyramidal cells and to altered expression of genes involved in the transcriptional regulation of synaptic genes, particularly the NMDA receptor subunit NR2A. They propose that the two glycosylases do not participate in DNA repair but instead regulate synaptic proteins.

The authors provided compelling evidence for altered behavior in the double KO mice compared to that of WT. However, for this Reviewer, the underlying mechanisms responsible for such alteration remains unclear. Particularly, the lack of correspondence between the behavioral, LTP and NMDA receptors subunits data is problematic. This may be due, in part, to the lack of detailed description of the methods and data presented.

1) In Methods, the detailed description of electrophysiological recordings is lacking which makes very difficult to follow the results and the interpretation of data. The exact position of the stimulating and recording electrodes should be indicated for each set of experiments. Are the recording electrodes located at the apical and basal dendrites of the pyramidal neurons? Where in the stratum oriens was the stimulating electrode positioned? Please also include the size (resistance) of the glass electrodes, the instruments (stimulator, amplifier, software) used, and the cutoff frequencies of filtered signals.

2) Explain why behavioral studies were done on 6-month-old male mice while electrophysiological studies were performed on 4-month-old male and female mice. Clearly, age and sex are important variables affecting synaptic plasticity and behavior.

3) Evidence for the reduced number/density of fibers hypothesized to account for the differences in prevoilley observed between WT and KO mice (Fig. 3A vi) should be provided. Explain how this finding would account for the results showing improved learning and memory (Fig.1 Morris water maze) in the absence of LTP alteration(Fig. 3B) and reduced N2RA/N2RB ratio (Fig. 5C).

4) Given that deletion of NEIL3 leads to cognitive impairment (Regnell et al., 2012 Cell Reports 2, 503–510) it would be nice to know whether the double (NEIL1/2) KO display compensatory changes in NEIL3 expression that could account for the improved learning and memory.

Other comments:

1) Figure 5A does not show GABRA-2 expression as the text says (P.8, l. 263).

2) Supplementary Fig. S5. The title of figure legend states "Reduced expression of GABRA and PSD95 in CA1", however, no significant ($p < 0.05$) change in PSD95 was detected ($p = 0.0762$ and $p = 0.0983$). Please correct the text.

3) Supplementary Fig. S3. The figure legend states "Black horizontal bar along the abscissa indicate $p < 0.05$ " but this bar is not shown in the figure.

REBUTTAL

Hildrestrand et al.:

NEIL1 and NEIL2 DNA glycosylases modulate anxiety and learning in a cooperative manner

We would like to thank the Reviewers for constructive criticism of our manuscript. The comments are highly judicious and we have strived to address each one. In general, changes have been made throughout the manuscript to meet the suggestions and comments made by the reviewers (highlighted in the revised manuscript). In particular, we have moderated our interpretations and conclusions. In cases where we have made substantial changes to the text, we refer to sections instead of line numbers. Notably, we have made some changes to the figures. In Figure 1, panels B-E and F-I now represent OF and EZM, respectively. Figure 2G has been changed to 2E (former E and F are now F and G). In supplementary information, two new figures have been added as Figures S4-2 and S5. Former Figures S4 and S5 have been changed to S4-1 and S6, respectively. Figures 2 A and B, 5, S2 and S5 have been modified to include individual data points. Please find our response below.

Best regards,

Magnar Bjørås

Reviewer #1 (Remarks to the Author):

The study of Hildrestrand et al., follows a number of studies that document phenotypes of mice deficient in DNA glycosylases operating in base excision repair. The mice have no/mild phenotypes, presumably because of functional overlap of different glycosylases. The main novelty of this paper, is the analysis of Neil1 Neil2 double knockout mice. The data from Neil1-KO mice are consistent with previous findings: increased weight starting in adulthood reflecting a mild metabolic syndrome (Vartanian, PNAS 2005); increased mobility in an open field test. The weight gain phenotype did not occur in Neil2-KO and double mutants, while double mutants showed a further increase in spontaneous mobility in open field as well as in an elevated maze. In addition, this study documents subtle changes in Morris water maze tests, hippocampal slice electrophysiology and hippocampal CA1 transcriptome, and no change in oxidative DNA damage markers. The data are relevant in that they reproduce previous findings, showing that something is going on in Neil1-KO mice and that this phenotype is modified by the absence of Neil2, and is not obviously linked to the accumulation of DNA damage. A major problem is that data are overinterpreted.

Specific comments:

1) Title, abstract and rest of the paper: Overall the authors have a tendency to oversell the data; I would strongly recommend to tone down the conclusions and present the data as they are. Statements like 'NEIL1 promotes spatial memory retention' (Canugovi et al., PNAS 2012) are misleading and disorienting young scientist. What is shown in that paper is that Neil1-KO mice display subtle changes in Morris water maze performance that are difficult to interpret and that by themselves are insufficient to draw any conclusions about a role of NEIL1 in spatial memory retention. Nevertheless, based on that title, NEIL1 is sold as a learning and memory factor in the literature, and perhaps also has distracted some of the scientists contributing to this paper.

Pleas tone down the title, abstract, and rest of the paper!! There is no data about learning and anxiety in the manuscript, nor is there evidence for a direct role of NEIL1 nor NEIL2 in hippocampal neurons. As a starter, statements about altered 'anxiety' in the open field require changes in the time spent in the center area, (eventually complemented by other parameters such as fecal boli deposits, etc). However, time in the center area is unaltered in mutants (Fig. 1H). Thus, basically the data indicate that, if anything, the mutants show

unaltered anxiety status. Likewise, the MWM data do not show changes in key parameters for learning and memory consistent with unaltered LTP. The data suggest altered time spend in specific non-target quadrants. These data are difficult to interpret, and can not be linked to memory performance.

Changes in behavior in KO mice (in particular global KO mice) do not necessarily uncover the function of the eliminated protein. Since elimination of NEIL1 alters the metabolic and perhaps the health status of the animals, this may influence behavioral readouts. In any case, the data do not support statements such as 'NEIL1 and NEIL2 regulate ...' (title) and 'NEIL1 and NEIL3 alter cognition ..' (abstract). Please, be more precise: 'the absence of NEIL1 and NEIL2 affectsbehavior, gene expression etc....' Also be more explicit about readouts that do not changes, rather than overselling marginal changes in non-relevant and poorly documented changes such as 'reduced stratum oriens afferents...'

We agree that there is no indication of less anxiety in the DKO mice in the OF test and that is why we did not refer to Figure 1H when presenting the anxiety data. We agree that the text (Results, lines 97-100) may be misinterpreted and adjustments have been made to clarify this issue in the revised version (see Results, behavior). Regarding the EZM test, our data indicate less anxiety in the DKO mice, as significantly more time spent in the open areas is a measure of less anxiety in the EZM¹⁻⁴. For clarity, Fig. 1 has been modified so that panels B-E represent OF and panels F-G represent EZM. Regarding the MWM test, the data clearly show that the DKO mice spend less time searching for the hidden platform compared to the WT mice during training (Figure 1J). The difference is statistically significant and is indicative of improved spatial learning ability⁵⁻⁷. Regarding memory testing in the MWM (d5 and d12), we agree that the data are less clear. We mention a tendency of the DKO mice to be less decisive on d12 vs. d5. However, since none of the measures (d5 and d12) were statistically significant, no conclusions on memory were made. The text has been modified to make this clear (see Results). The *Neil1*^{-/-} and *Neil2*^{-/-} mice showed similar behavior, regardless of the weight differences, and the *Neil1*^{-/-} mice did not show any signs of illness; thus, metabolic function and health status in *Neil1*^{-/-} mice are unlikely to have an impact on behavior. This has been discussed in more detail (see Discussion).

2) Figure 1 and 2

For behavioral experiments provide more relevant details in methods/results section. I have the impression that single and double mouse lines are bred independently and that

'controls' are not littermates, generated from mating of heterozygotes, which would be methodologically more correct. Are mice group housed during the experiments; are the same mice used for different experiments, and if so, what is the order of experiments. Is the same cohort used for DNA damage analysis?

As suggested by reviewer 1, we have not used breeding of heterozygous mice. Since the mutants were already backcrossed at least 8 times onto the C57BL/6N background when we started the project, we chose to breed the lines separately and use C57BL/6N as control mice. This is in agreement with the three Rs (Replacement, Reduction and Refinement) in animal research, since fewer mice are required to obtain enough KO and DKO mice than when using het x het breeding. The mice were housed with their littermates (max 5 in each cage). The same mice were tested in the three mazes, however; since the OF and EZM are less time consuming than the MWM, more mice were included in these two tests. The test order was as follows: OF, EZM and MWM. The mice tested in the MWM were weighed after the last probe test. New cohorts of mice were used for DNA damage analysis and all other experiments included in this study.

Additional information regarding breeding protocols and behavioral studies has been included in the Methods section.

3) Figure 4 and S4

RNAseq in combination with laser dissection is more noisy than other approaches, and the experiments seemingly was poorly designed resulting in insufficient samples of Neil1 and double KO mice. This may explain more significant changes in Neil2-KO. Is there a PC-plot without the 'outliers'?

In PC-plot in Fig. S4G use different symbols (in addition to colors) to indicate different mouse types. Tone down title of this section: the DEG have a very putative, indirect link with synaptic function: just state that Neil1 and Neil2 KO have a differential effect on CA1 gene expression

In the study we present here, we chose a highly region-specific tissue isolation approach that allows for a selective analysis of CA1 pyramidal cells. The laser-dissection method we used (see also "Methods") requires a considerable investment of resources, both with respect to time and material. We agree with the reviewer that a higher number of individuals per group would improve the quality of the dataset. Yet, including a significantly higher number

of individuals is well beyond the scope and possibilities even of a well-equipped lab. Nonetheless, we chose the laser-dissection method as other methods such as manual tissue isolation followed by FACS-sorting cannot differentiate between e.g. dorsal and ventral hippocampal tissue. The transcriptional differences alongside the dorsoventral axis (see e.g. Soltesz and Losonczy⁸) and also between different hippocampal subfields is so considerable that we regarded this as a potentially devastating confounder warranting a highly region-specific approach over a higher number of individuals using less specific material. An additional PCA plot without the outliers has been included as Figure S4-2 in the supplementary information and former Figure S4 has therefore been changed to Figure S4-1 (see below). According to the reviewer's suggestion, the titles of the RNA seq section and Figure S4-1 have been toned down and we added symbols in addition to colors to the PCA plots.

Figure S4-1 G: PCA plot, all samples included

Figure S4-2: PCA plot without outliers

4) Figure 5 and S5

These data are fully inconclusive: Synapses of CA1 pyramidal neurons localize in the stratum radiatum, moleculare and oriens rather than in the soma of the pyramidal neurons. What is stained here is not synaptic proteins. Please use Western blot to demonstrate changes in NMDA and GABA receptor status There are also problems with the reasoning. If somehow Neil2 via Nr1d1 etc. regulates synapses this also would result in changes in receptor mRNA's.

We would like to thank the reviewer for this comment. While it is correct that the stratum oriens and radiatum are the crucial “relais” areas for synaptic input to CA1, we need to point out that our data, as shown in Fig. 5, also refers to these regions specifically. We see that the differences observed in stratum pyramidale are equally present in stratum oriens and stratum radiatum with respect to the NR2A/B ratio (Fig. 5C) and, in part, for absolute differences in NR2A and B subunits (Fig. 5A and B). Further, we would like to point out that we disagree with the reviewer’s conclusion that the immunohistochemical reactivity observed across stratum pyramidale must be artefact or “not synaptic proteins”. Studies by Dodt et al⁹ and Petralia et al¹⁰ show that the synaptic subunits we examined are equally present across the stratum pyramidale as well as radiatum and oriens, particularly at the

proximal dendrite. This also explains why we see a somewhat higher reactivity across the stratum pyramidale even though the somatic fraction of the examined subunits is not necessarily the most abundant one. Here, a lot of proximal dendrites arise from the somata of the stratum pyramidale and overlap. The further distal we go from the stratum pyramidale, the less dendrites overlap, leading to a less impressive reactivity pattern. While our line of reasoning would have been greatly supported by differentially regulated receptor subunits in the RNA seq analysis, we still regard our argumentation as valid: It is known that receptor composition is a highly plastic, quick and activity-driven process involving multiple levels of regulation going well beyond mere differential transcription (see e.g. Paoletti et al¹¹). Regarding the western blot technique, it does not allow for quantification in a context-specific manner, as it doesn't maintain the cellular characteristics and structure of the tissue. In addition, the epitopes recognized by the primary antibodies in IHC may not be identically available in WB.

5) The data on effects of Neil KO on weight are not discussed. What happens in double KO compared to Neil1 KO?

The weight phenotype has been discussed in more detail (see Results and Discussion). Inactivation of *Neil2* in addition to *Neil1* seems to rescue the weight phenotype observed in the NEIL1 KO mice. Alternatively, it could be a result of the variable penetrance of the *Neil1*^{-/-} metabolic phenotype¹².

Interestingly, NR1D1 and NR4A orphan receptors, which are differentially regulated only in NEIL2- and NEIL1/NEIL2-deficient mice have been shown to be involved in glucose metabolism^{13,14} and lipid metabolism^{15,16}. However, changes in the transcriptome in the CA1 region might not have an immediate relevance for metabolic regulation and conclusion cannot be drawn. Since metabolism is not the focus of this study, we have not conducted any additional experiments to look further into this, but the weight phenotype is certainly an observation that warrants additional investigation.

Reviewer #2 (Remarks to the Author):

NEIL1 and NEIL2 are enzymes which display glycosylase activity and take part in the initial steps of the base excision repair (BER) pathway. Several studies suggest that BER plays an important role in protecting the brain from oxidative DNA damage. In this manuscript by Hildestrand et al., the authors investigate a possible non-canonical function of NEIL1 and NEIL2 in modulating memory formation and anxiety in mice. Based on their observations they postulate that within CA1 of the hippocampus NEIL1 and NEIL2 jointly regulate transcription of genes relevant for synaptic function. Overall, this is an interesting report which sheds light on prospective alternative roles of the NEIL1/2 glycosylases in brain in addition to their roles in the canonical BER pathway. The mechanism behind this function is, however, not fully elucidated.

Overall the manuscript is well written and the experimental work is appropriately described. Nevertheless, some issues do need consideration as described below.

Comments:

1) Some of the results obtained in this study, including accumulation of DNA damage, may depend strongly on the age of the mice. Therefore, it is important that the age of the mice in this study is clearly stated in the Results section and that potential issues regarding age are discussed in more depth in the Discussion.

We have included more detailed information regarding the age of the animals in the Results section (according to what is also stated in the Methods section). The mice ranged from 4-6 months, but the majority of mice used were 6-month-old male mice.

2) The authors should discuss in more detail the difference between what is measured in the open field test and the elevated zero maze. This is an important issue since significant differences between WT and NEIL deficient mice are observed only in Fig 1I but not in Fig 1H. Accordingly, the authors should modify their statement in Line 278-280, where they claim that their study reveals reduced anxiety in the open field test and the elevated zero maze.

Differences between OF and EZM have been clarified in the manuscript (Results section) and Fig. 1 has been modified so that panels B-E represent OF and panels F-I represent EZM. We agree that there is no indication of less anxiety in the DKO mice in the OF test. Hence, we did not refer to Figure 1H when presenting the anxiety data. We agree that the text (Results, lines 97-100) is open to misinterpretation and adjustments have been made to clarify this issue in the revised version. In the EZM, the principle variable measured as an anxiety marker is the percentage of time spent in the open areas, where less time spent in the open areas suggests increased anxiety^{2,4}. Thus, our EZM data indicate reduced anxiety in the DKO mice, as the mice spent significantly more time in the open areas than the single mutants and WT mice.

3) In line 107-109 it says: during the 2nd retention trial (day 12), the NEIL1-/-NEIL2-/- mice became less decisive, shown as decreased preference for the target zone (Fig 1K). The authors should clearly state whether this is just a trend or statistically significant and the observed differences between day 4, 5 and 12 should be discussed in more detail.

We mentioned a tendency of the DKO mice to be less decisive on d12 vs. d5 in the Results section. However, since none of the measures (d5 and d12) were statistically significant, no conclusions on memory were made. The text (Results section) has been altered to make this clear.

4) The authors suggest that the observed changed behaviour in the KO mice may be due to a role for NEIL1/NEIL2 in epigenetic gene regulation. In order to test this hypothesis, the authors should consider to compare the methylation pattern for the Npas4 and Nr4a1-3 gene loci (or other DEGs), respectively, in the WT and the KO mice.

We appreciate the suggestion and this is something that we have planned to do down the line. For now, we will follow the Editor's conclusion on this point:

Editor: "While we agree with Referee #2 that investigating an epigenetic role for Neil1/2 would improve the impact of this study, we would not consider this point as necessary for consideration of a revised manuscript."

Comments for specific figures:

Figure 1 A: The authors should consider to revise the illustration of the elevated zero maze – the perspective of the current figure does not seem to be correct

This has been corrected.

Figure 1 J-K: The figure legend lacks information regarding colour code (shades of grey and black) for the round symbols.

Round symbols indicate individual mice; outliers in black and all others in grey. Different shades of grey are caused by overlapping mice. This information has been added to the figure legend.

Figure 2B: The authors should comment on why they have chosen to focus on 5-ohC and not e.g. 5-ohU when estimating damage accumulation. Is one type of adduct more common or a better substrate for NEIL1 or NEIL2 than the other, or?

Both 5-ohC and 5-ohU are substrates for NEIL1 and NEIL2. However, measuring 5-ohC by HPLC-MS/MS is much easier and more reliable than measuring 5-ohU, thus, we chose to focus on 5-ohC.

Figure 2 C: From reading the Materials and Methods section and the figure legend it is not clear what would be equal to 100% comet tail intensity. Also, it does not seem obvious, why the tail intensity appears to be lower and not higher after treatment with Fpg. Please clarify.

The comet assay detects strand breaks in the DNA of a cell presented as comet tail intensity. The DNA damage levels (tail intensities) observed were recorded as appearing in the software, under the experimental conditions described with respect to gel concentration and electrophoresis conditions. A cell exhibiting 0% tail intensity has no DNA in the tail, and hence no detectable DNA damage under the conditions used. Increasing fluorescence in the tail vs the head of the comet indicates increasing DNA damage levels, up to a level of 100%, where the entire DNA is present in the tail. This information has been added to the Methods section.

Our Comet data revealed two things:

- 1) The strand break levels in Fpg-untreated cells are the same in WT and NEIL1 KO samples, but reduced in NEIL2 KO and NEIL1/NEIL2 DKO samples.
- 2) Tail intensities were unchanged (not lower) in all samples after treatment with Fpg.

Regarding 1, we speculate that NEIL2 could be involved in making the DNA more accessible to strand breaks, since the absence of NEIL2 caused less strand breaks. However, in order to understand the molecular basis of this intriguing result, further investigation is needed.

Regarding 2, treatment with Fpg normally leads to increased tail intensities caused by additional strand breaks generated during removal of unrepaired base lesions. Our data suggests that the amount of base lesions giving rise to more strand breaks after treatment with Fpg is not significantly changing tail intensity, in any of the samples/genotypes.

In conclusion, the mutants had not accumulated significantly more strand breaks (Fpg-untreated) or base lesions (Fpg-treated) in hippocampus than the WT mice.

Figure 2 D: The 600 SNPs, which are present in the double KO, but not in the single KO mice – are these evenly distributed in different regions of the genome?

Yes, as demonstrated in fig. 2F (previously 2E) the distribution of all variants across genomic regions is similar between the three mutants. This calculation is based on the total number of variants in each mutant.

Figure 2 E: Previous studies of NEIL1 and NEIL2 point to a potential role for these enzymes in transcription associated DNA repair. Therefore, it would be interesting to also present the data from the current study in a way, where the genes are separated into expressed and non-expressed genes, respectively, for each of the genotypes.

We thank the reviewer for the suggestion. Although it is not possible to look at non-expressed genes in a RNA seq dataset, we have chosen to categorize all sequenced genes based on the normalized counts into a lower (25%) and upper (75%) quartile, representing low- and high-expressed genes, respectively. Interestingly, we found an enrichment of DEGs from the NEIL-deficient mice in the high-expressed gene category (NEIL 1: 48%, NEIL 2: 41%, NEIL 1/2: 45%) whereas less than 1.5% of DEGs are found in the low expressed gene category for each mutant mice. These results support previous *in vitro* findings that NEIL DNA glycosylases may preferentially affect transcriptionally active genes.

We have included the findings as Figure S5 (see below) and referred to it in the results and discussion section of the revised manuscript.

Supplementary Figure S5. NEIL-deficient mice alter preferentially high-expressed genes. Normalized counts (in log2 scale) of all sequenced genes (upper panel) and the DEGs (lower panel) in *Neil1*^{-/-} (A), *Neil2*^{-/-} (B), *Neil1*^{-/-}*Neil2*^{-/-} (C) and wildtype (D) mice were shown in histograms. The 25% lower quartile and 75% upper quartile (based on all sequenced genes, indicated by the red line) were used as criteria to define the low- and high-expressed genes, respectively.

Figure 2 G: Indication of colour code (blue, green and purple) is missing

Panel G in the submitted figure has been changed to panel E for clarity and the missing color code has been included. Former panels E and F are now F and G, respectively.

Figure 4 B: It looks as if the difference in e.g. total # DEGs between NEIL1 ^{-/-} and NEIL2 ^{-/-} is significant – but why does this diagram does contain any indications of significance?

Differentially expressed genes were identified based on the pooled data of several mice, see also fig. S4-1 G and H. This type of analysis does not allow for a display of “indications of

significance". All DEGs were, on the other hand, significantly differentially expressed, see also "Methods".

Minor points:

In several cases periods are lacking at the end of sentences (after a reference index number). This has been corrected (not highlighted in the manuscript since corrections were made in several places).

The layout for the references is not the same throughout the manuscript
This has been corrected.

Abbreviations are not always introduced or explained first time they are used
This has been corrected.

Line 36: the authors should consider to write " ..., while the role of NEIL2 remains unclear"
This has been changed.

Line 77-78: the authors should consider to write a few more words to explain what is meant by "mild metabolic phenotype"
"Mild" has been removed and "with variable penetrance" and the reference Sampath et al, 2011, have been added.

Line 95-96: the authors should consider to write "by being more mobile" instead of "by being more active", since they initiate the sentence by writing "mice displayed hyperactive behaviour"
This has been changed.

Line 112: the authors should also comment on the observed significant differences between the two single ko mouse genotypes and between the NEIL1 -/- and NEIL1-NEIL2 double ko mice shown in Figure S1
The differences have been commented upon in the Results section and discussed in the Discussion section.

Line 133-136: it would be appropriate to emphasize that for the data presented in Figure S2 the level of damage is measured in a single specific gene and not in the whole genome
“in the *Gapdh* gene, which is transcriptionally active and thus, easily accessible for damage induction, but also repair.” has been added, both to the Results section and to Methods.

Line 136: “hippocampus” should be “hippocampi”
This has been changed.

Line 138: Figure S4 is mentioned before Figure S3 (Line 195)
The reason for this is that the RNA seq data were discussed in a later section (lines 207-239), but we referred to Figure S4 in the DNA damage section (line 138) to show that there was no compensatory upregulation of other oxidative DNA glycosylases that could explain the lack of accumulation of DNA lesions in the mutants. The sentence regarding RNA seq and DNA glycosylases (lines 137-149) has been moved to the RNA seq section in the Results section.

Line 140: “suggest” should be “suggests”
This has been changed.

Line 152 and 154: the text refers to Figure 4E and 4F. Do they mean 2E and 2F?
Yes, this has been corrected. However, since the figure has been modified we refer to 2F and 2G.

Line 167 and 170: the authors should consider to write “intensities” instead of “strengths”
This has been changed.

Line 251: abbreviation instead of full-length word for “stratum pyramidale” should be used as the abbreviation has been introduced earlier
This has been corrected.

Line 253, 359 and 362: abbreviation instead of full-length word for “stratum oriens” should be used as the abbreviation has been introduced earlier
This has been corrected.

Line 281: in this line it is not completely clear whether the authors refer the NEIL1/2 deficient mice or the OGG1/MUTYH mice, therefore this sentence should be edited for clarification.

This has been corrected.

Line 317-318: the authors should elaborate on their sentence “we may speculate that this is caused by a putative role of NEIL2 in processes making the chromatin more accessible to strand breaks”

Open chromatin is more sensitive to strand breaks than heterochromatin. If NEIL2 plays a role in chromatin modulation, the NEIL2-deficient mice may contain more heterochromatin. However, this is speculation and merits further investigation.

Line 451: “using following” should be “using the following”

This has been corrected.

Line 668: the authors should consider to use the word “sacrificed” instead of “killed”

This has been corrected.

Reviewer #3 (Remarks to the Author):

DNA glycosylases play important roles in the repair of DNA resultant from oxidative damage and thus participate in the prevention of neuronal death after ischemia and age-related cognitive decline. The present manuscript describes the effects of deletion of two DNA glycosylases (NEIL1 and NEIL2) on the behavior of single and double knockout (KO) mice and evaluated potential underlying mechanisms.

Previous work by this group have provided evidence that deletion of the DNA glycosylase NEIL3 resulted in impaired learning and memory and reduced anxiety levels which was attributed to reduced proliferative capacity of neural stem cells due to impaired DNA repair capacity. They now report that mice lacking both NEIL1 and NEIL2 glycosylases do not show impaired DNA oxidative repair mechanism but exhibited reduced anxiety and improved learning and memory compared to control wild-type (WT) mice. These behavioral alterations were attribute to reduced hippocampal stratum oriens afferents into pyramidal cells and to altered expression of genes involved in the transcriptional regulation of synaptic genes, particularly the NMDA receptor subunit NR2A. They propose that the two glycosylases do not participate in DNA repair but instead regulate synaptic proteins.

The authors provided compelling evidence for altered behavior in the double KO mice compared to that of WT. However, for this Reviewer, the underlying mechanisms responsible for such alteration remains unclear. Particularly, the lack of correspondence between the behavioral, LTP and NMDA receptors subunits data is problematic. This may be due, in part, to the lack of detailed description of the methods and data presented.

1) In Methods, the detailed description of electrophysiological recordings is lacking which makes very difficult to follow the results and the interpretation of data. The exact position of the stimulating and recording electrodes should be indicated for each set of experiments. Are the recording electrodes located at the apical and basal dendrites of the pyramidal neurons? Where in the stratum orien was the stimulating electrode positioned? Please also include the size (resistance) of the glass electrodes, the instruments (stimulator, amplifier, software) used, and the cutoff frequencies of filtered signals.

Requested information has been added to the revised manuscript (Methods, Electrophysiology).

2) Explain why behavioral studies were done on 6-month-old male mice while electrophysiological studies were performed on 4-month-old male and female mice. Clearly, age and sex are important variables affecting synaptic plasticity and behavior.

Although we agree that using mice of the exact same age in different experiments is optimal, this is not always feasible. According to a study comparing different behavioral tasks in aging C57BL/6 mice, the middle age groups (4-5 mo and 6-7 mo) showed similar behavior in the open field and the elevated plus maze tests¹⁷. Thus, comparing mice that range from 4 to 6 months appeared justifiable.

The use of both female and male mice was based on previous observations showing that the LTP level was indistinguishable between the genders. However, the authors agree that optimally only males should have been used.

3) Evidence for the reduced number/density of fibers hypothesized to account for the differences in prevoiley observed between WT and KO mice (Fig. 3A vi) should be provided. Explain how this finding would account for the results showing improved learning and memory (Fig.1 Morris water maze) in the absence of LTP alteration (Fig. 3B) and reduced N2RA/N2RB ratio (Fig. 5C).

In our manuscript we provide evidence that the absolute amount of NR2A-receptor subunits and NR2A/B ratio in SO of *Neil1*^{-/-}*Neil2*^{-/-} mice are significantly reduced (Fig. 5A and C). As we state both in the results and discussion part, the NR2A-subunit is a crucial functional subunit in the excitatory NMDA-receptor complex and, in our opinion, given the changed NR2A/B ratio, provides immunohistochemical evidence for an altered receptor composition that may help explain the electrophysiological findings. In addition, we find a tendency of lower PSD-95 reactivity in NEIL1/NEIL2-deficient CA1 (Fig. S6B). Even though the latter refers to SP only due to technical reasons, SP reactivity reflects overlapping proximal dendrites as well, as stated in the answers to reviewer 2 above. Consequently, these data can point to an overall lower amount of afferent fibers. In the results part, we adjusted our statement of fiber density, adding that our electrophysiological findings can also be explained by a differential receptor composition.

Regarding LTP, the coupling between LTP in CA1 and spatial reference memory in the MWM has been questioned¹⁸⁻²⁰. Thus, the unaltered LTP in the *Neil1*^{-/-}*Neil2*^{-/-} mice does not

necessarily contradict the improved learning.

4) Given that deletion of NEIL3 leads to cognitive impairment (Regnell et al., 2012 Cell Reports 2, 503–510) it would be nice to know whether the double (NEIL1/2) KO display compensatory changes in NEIL3 expression that could account for the improved learning and memory.

We observed no compensatory upregulation of NEIL3 in hippocampus (CA1) of the NEIL1/NEIL2 DKO mice. This was mentioned in the Results / DNA damage detection part (lines 137-140) in the submitted manuscript and has been moved to the Results / RNA seq data section in the revised manuscript.

Other comments:

1) Figure 5A does not show GABRA-2 expression as the text says (P.8, l. 263).

The text could be misinterpreted and has been changed to clarify the figure references.

2) Supplementary Fig. S5. The title of figure legend states “Reduced expression of GABRA and PSD95 in CA1”, however, no significant ($p < 0.05$) change in PSD95 was detected ($p = 0.0762$ and $p = 0.0983$). Please correct the text.

PSD-95 has been removed from the title of the figure legend (Fig. S6 in revised version).

3) Supplementary Fig. S3. The figure legend states “Black horizontal bar along the abscissa indicate $p < 0.05$ ” but this bar is not shown in the figure.

The sentence has been removed from the figure legend as there is no significant difference between the genotypes.

References

1. Shepherd, J. K., Grewal, S. S., Fletcher, A., Bill, D. J. & Dourish, C. T. Behavioural and pharmacological characterisation of the elevated "zero-maze" as an animal model of anxiety. *Psychopharmacology (Berl)*. **116**, 56–64 (1994).
2. Kulkarni, S. K., Singh, K. & Bishnoi, M. Elevated zero maze: A paradigm to evaluate antianxiety effects of drugs. *Methods Find. Exp. Clin. Pharmacol.* **29**, 343–348 (2007).
3. Sestakova, N., Puzserova, A., Kluknavsky, M. & Bernatova, I. Determination of motor activity and anxiety-related behaviour in rodents: Methodological aspects and role of nitric oxide. *Interdisciplinary Toxicology* **6**, 126–135 (2013).
4. Tucker, L. B. & McCabe, J. T. Behavior of male and female C57Bl/6J mice is more consistent with repeated trials in the elevated zero maze than in the elevated plus maze. *Front. Behav. Neurosci.* **11**, (2017).
5. Morris, R. Developments of a water-maze procedure for studying spatial learning in the rat. *J. Neurosci. Methods* **11**, 47–60 (1984).
6. Vorhees, C. V & Williams, M. T. Morris water maze: procedures for assessing spatial and related forms of learning and memory. *Nat. Protoc.* **1**, 848–858 (2006).
7. Barnhart, C. D., Yang, D. & Lein, P. J. Using the Morris water maze to assess spatial learning and memory in weanling mice. *PLoS One* **10**, (2015).
8. Soltesz, I. & Losonczy, A. CA1 pyramidal cell diversity enabling parallel information processing in the hippocampus. *Nature Neuroscience* **21**, 484–493 (2018).
9. Dodt, H. U. NMDA and AMPA receptors on neocortical neurons are differentially distributed. *Eur. J. Neurosci.* **10**, 3351–3357 (1998).
10. Petralia, R. S., Wang, Y. X. & Wenthold, R. J. The NMDA receptor subunits NR2A and NR2B show histological and ultrastructural localization patterns similar to those of NR1. *J. Neurosci.* **14**, 6102–6120 (1994).
11. Paoletti, P., Bellone, C. & Zhou, Q. NMDA receptor subunit diversity: impact on receptor properties, synaptic plasticity and disease. *Nat. Rev. Neurosci.* **14**, 383–400 (2013).
12. Sampath, H. *et al.* Variable penetrance of metabolic phenotypes and development of high-fat diet-induced adiposity in NEIL1-deficient mice. *Am. J. Physiol. Metab.* **300**, E724–E734 (2011).
13. Pei, L. *et al.* NR4A orphan nuclear receptors are transcriptional regulators of hepatic glucose metabolism. *Nat. Med.* **12**, 1048–1055 (2006).
14. Fleming, A. M., Zhu, J., Howpay Manage, S. A. & Burrows, C. J. Human NEIL3 Gene Expression Regulated by Epigenetic-Like Oxidative DNA Modification . *J. Am. Chem. Soc.* **141**, 11036–11049 (2019).
15. Zhao, Y. & Bruemmer, D. NR4A orphan nuclear receptors: Transcriptional regulators of gene expression in metabolism and vascular biology. *Arteriosclerosis, Thrombosis, and Vascular Biology* **30**, 1535–1541 (2010).
16. Reinke, H. & Asher, G. Circadian Clock Control of Liver Metabolic Functions. *Gastroenterology* **150**, 574–580 (2016).
17. Shoji, H., Takao, K., Hattori, S. & Miyakawa, T. Age-related changes in behavior in C57BL/6J mice from young adulthood to middle age. *Mol. Brain* **9**, 11 (2016).
18. Zamanillo, D. *et al.* Importance of AMPA receptors for hippocampal synaptic plasticity but not for spatial learning. *Science (80-.)*. **284**, 1805–1811 (1999).

19. Bannerman, D. M. *et al.* Dissecting spatial knowledge from spatial choice by hippocampal NMDA receptor deletion. *Nat. Neurosci.* **15**, 1153–1159 (2012).
20. Bannerman, D. M. *et al.* Hippocampal synaptic plasticity, spatial memory and anxiety. *Nat. Rev. Neurosci.* **15**, 181–192 (2014).

Reviewers' comments:

Reviewer #1 (Remarks to the Author):

COMMSBIO-21-0346-A revision

The manuscript has been improved in several ways. There are still some points that may need some attention:

1) Reviewer 1 point 2)

Author's reply: 'As suggested by reviewer 1, we have not used breeding of heterozygous mice. Since the mutants were already backcrossed at least 8 times onto the C57BL/6N background when we started the project, we chose to breed the lines separately and use C57BL/6N as control mice. This is in agreement with the three Rs (Replacement, Reduction and Refinement) in animal research, since fewer mice are required to obtain enough KO and DKO mice than when using het x het breeding. The mice were housed with their littermates (max 5 in each cage). The same mice were tested in the three mazes, however; since the OF and EZM are less time consuming than the MWM, more mice were included in these two tests. The test order was as follows: OF, EZM and MWM. The mice tested in the MWM were weighed after the last probe test. New cohorts of mice were used for DNA damage analysis and all other experiments included in this study.

Additional information regarding breeding protocols and behavioral studies has been included in the Methods section.

****: The statement about 3R's is not convincing. A bit of extra breeding for methodological reasons would have been defensible. The fact that behavioral data are obtained from lines bred separately, requires explicit discussion in the discussion section.

2) Reviewer 1 point 3)

Author's reply: In the study we present here, we chose a highly region-specific tissue isolation approach that allows for a selective analysis of CA1 pyramidal cells. The laser-dissection method we used (see also "Methods") requires a considerable investment of resources, both with respect to time and material. We agree with the reviewer that a higher number of individuals per group would improve the quality of the dataset. Yet, including a significantly higher number RNAseq of individuals is well beyond the scope and possibilities even of a well-equipped lab.

****: Are the authors suggesting that it is valid to publish datasets of insufficient quality when resources are limited?

3) Reviewer 1 point 4)

Author's reply: '..... Further, we would like to point out that we disagree with the reviewer's conclusion that the immunohistochemical reactivity observed across stratum pyramidale must be artefact or "not synaptic proteins". Studies by Dodt et al. and Petralia et al. show that the synaptic subunits we examined are equally present across the stratum pyramidale as well as radiatum and oriens, particularly at the proximal dendrite. This also explains why we see a somewhat higher reactivity across the stratum pyramidale even though the somatic fraction of the examined subunits is not necessarily the most abundant one. Here, a lot of proximal dendrites arise from the somata of the stratum pyramidale and overlap. The further distal we go from the stratum pyramidale, the less dendrites overlap, leading to a less impressive reactivity pattern. While our line of reasoning would have been greatly supported by differentially regulated receptor subunits in the RNA seq analysis, we still regard our argumentation as valid: It is known that receptor composition is a highly plastic, quick and activity-driven process involving multiple levels of regulation going well beyond mere differential transcription (see e.g. Paoletti et al11). Regarding the western blot technique, it does not allow for

quantification in a context specific manner, as it doesn't maintain the cellular characteristics and structure of the tissue.

In addition, the epitopes recognized by the primary antibodies in IHC may not be identically available in WB.

**** Immunohistological analysis of post-synaptic receptors, in particular NMDA receptors is not trivial. For example see Schneider Gasser 2006 Nat Prot (doi:10.1038/nprot.2006.265): .." loss of antigenicity and/or epitope masking that are due to tissue fixation are major limitations of immunohistochemistry, notably for the detection of postsynaptic proteins. Aldehyde fixation may alter antigenic sites by cross-linking tissue proteins and antibodies into the postsynaptic density and into the synaptic cleft, a narrow space that separates the presynaptic and postsynaptic elements and is filled with extracellular matrix molecule ..". Cytoplasmic NR2A and B staining in the cell body/ prox. dendrite of PFA perfusion fixed brain largely represents a non-synaptic pool of receptors. Instead milder fixation procedures yield synaptic staining which is concentrated in the str. rad and str or (see Table 1 of Schneider Gasser 2006; and Fritchey et al J Comp Neurol, 1998, 390:194-210).

It is still not clear what we see in the images of Fig. 5, and how it is connected to synaptic NMDA receptors. Does every dot reflect a psd with NMDA receptors? According to the Methods the images reflect 'a background-filtered signal indicating the approximate portion detected by the spot detection tool'. Please explain and show what type of 'background' has been filtered by the software. Is the dotted labeling visible before 'filtering'? How is 'background' defined?

Also explain what is meant by: "reactivity levels" based on Imaris spot detection tool in the bar graphs (Figure legends of Fig. 5; y-axis of bar graphs). Do values reflect relative signal intensity, number of labeled spots, or a combination of both.

Did the authors perform Western blot? There are many NR2A/B antibodies that work on Western. If the authors want to stick to their story that NR2A/B levels are altered, Western blot would be insightful.

Reviewer #2 (Remarks to the Author):

The authors have addressed the comments from this reviewer to a satisfactory level, and this reviewer does not have additional concerns regarding the revised manuscript.

Reviewer #3 (Remarks to the Author):

The authors have satisfactorily addressed some but not all my comments. Particularly remaining concern is the one regarding the discrepancy between the age and sex of the mice used for behavioral studies (6 months old male mice) vs those used for electrophysiological studies (4-6 months male and female mice), and the one regarding the meaning of results showing differences in stratum oriens prevoiley and the MWZ results in face of lack of changes in LTP.

They argued that in terms of behavioral tasks (open field and elevated plus maze) a report (Shoji et al., 2016) indicated that there is no difference in performance between 4-6 months old mice. Yes, this maybe the case but this was not my question since the authors only used 6 months old male mice for the behavioral studies (all the same age and sex). My question was related to the age and sex

discrepancies between the behavioral and electrophysiological studies. There are reports indicating age related changes in receptor density and calcium channels in the hippocampus (see Huang and Kandel, 2006 *Learning and Memory*, 13:298) and other reports on the influence of estrogen on synaptic plasticity and spatial learning (e.g., Wang et al., 2018 *J Neurosci* 38:7935; Yagi et al 2019 *Neuropharmacol* 44: 200). How can the authors compare electrophysiological data with the behavioral ones, if each one was performed in different age and sex cohorts? If the authors want to argue differently, please show that in your data set there is no sex and age differences. Alternatively, the possibility that the electrophysiological results maybe biased due to sex-dependent effect could be added to the discussion.

Added to this problem is the change in NMDA receptor composition reported by immunohistochemistry that was obtained from 3-6 months old male mice. How sure are the authors that the changes in NMDA receptor composition faithfully correlates with stratum oriens axonal activation (from females and males) and are not correlated with age and sex? How does this change in axonal activation affect the learning process, given the unaltered LTP, which has been attributed to a low NR2A/NR2B-ratio (as the authors mentioned line 268)? The authors speculates that the changes in axonal activation could be related to a decrease in density of afferents to CA1 that would be sufficient to improve spatial coding independently of increased LTP. Again, evidence that there is no sex and age difference should be provided or added in the discussion.

REBUTTAL # 2

Hildrestrand et al.:

NEIL1 and NEIL2 DNA glycosylases modulate anxiety and learning in a cooperative manner

We would like to thank the reviewers for pointing out issues in the revised manuscript that are still not clear. In the second revision we have strived to answer all the questions/comments made by the reviewers. Of note, the colors in figures 1, 2 and 3 have been adjusted after feedback from one of the coauthors who had difficulties separating some of the colors in the previous versions. Further, figure S3 has been excluded from the paper and thus, figures S4-1, S4-2 and S5 have been relabeled S3-1, S3-2 and S4, respectively. Figure 5 has been moved to Supplementary Information as figure S5. For details, see our response below.

Best regards,
Magnar Bjørås

Reviewer #1 (Remarks to the Author):

COMMSBIO-21-0346-A revision

The manuscript has been improved in several ways. There are still some points that may need some attention:

1) Reviewer 1 point 2)

Author's reply: 'As suggested by reviewer 1, we have not used breeding of heterozygous mice. Since the mutants were already backcrossed at least 8 times onto the C57BL/6N background when we started the project, we chose to breed the lines separately and use C57BL/6N as control mice. This is in agreement with the three Rs (Replacement, Reduction and Refinement) in animal research, since fewer mice are required to obtain enough KO and DKO mice than when using het x het breeding. The mice were housed with their littermates (max 5 in each cage). The same mice were tested in the three mazes, however; since the OF and EZM are less time consuming than the MWM, more mice were included in these two tests. The test order was as follows: OF, EZM and MWM. The mice tested in the MWM were weighed after the last probe test. New cohorts of mice were used for DNA damage analysis and all other experiments included in this study.

Additional information regarding breeding protocols and behavioral studies has been included in the Methods section.

****: The statement about 3R's is not convincing. A bit of extra breeding for methodological reasons would have been defensible. The fact that behavioral data are obtained from lines bred separately, requires explicit discussion in the discussion session.

To clarify what we mean, we have made a simple calculation:

To get both single and double KO mice that are siblings, one would have to breed double heterozygous mice. In theory, 6.25% of the pups will be either WT, single KOs or double KO. Thus, if we expect an average of 7 pups per litter, one would need at least 54 breedings, giving a total of 378 pups, to obtain 12 WT, 12 NEIL1 KO, 12 NEIL2 KO and 12 NEIL1/NEIL2 DKO male mice ($(378 \times 0,0625) / 2$ sexes). In contrast, breeding of separate (backcrossed) lines requires 4 breedings of each genotype, giving a total of 112 pups ($4 \times 4 \times 7$), to obtain 14 males of each genotype ($112 / 4$ genotypes / 2 sexes).

Based on these calculations, it is fair to argue that using separate lines highly reduces the number of mice needed and thus, is in agreement with "Reduction". We would like to point out that all lines

were backcrossed for at least 8 times onto the same background as control mice and that we do not believe that breeding separate lines is valid if the mice are not backcrossed. See also: <https://www.jax.org/jax-mice-and-services/customer-support/technical-support/breeding-and-husbandry-support/considerations-for-choosing-controls>

Information regarding breeding of separate lines has been included both in the Methods and in the Discussion sections.

2) Reviewer 1 point 3)

Author's reply: In the study we present here, we chose a highly region-specific tissue isolation approach that allows for a selective analysis of CA1 pyramidal cells. The laser-dissection method we used (see also "Methods") requires a considerable investment of resources, both with respect to time and material. We agree with the reviewer that a higher number of individuals per group would improve the quality of the dataset. Yet, including a significantly higher number RNAseq of individuals is well beyond the scope and possibilities even of a well-equipped lab.

****: Are the authors suggesting that it is valid to publish datasets of insufficient quality when resources are limited?

We deeply regret if our previous line of argumentation in the earlier rebuttal letter created the notion of us supporting low-quality datasets to be published when resources are limited. This was not and never will be our attitude towards good scientific practice. On the contrary, we still think that our dataset is of good quality and technical integrity, as we tried to explain in our previous reply. We do agree, however, that a higher number of individuals would have been preferable as we already agreed and stated in the first answer to the reviewer.

At this stage, it might be worth to point out that many studies try to inflate "n" by counting e.g. single dissectates, single cells or synapses instead of animals, which is not valid due to the nested data problem¹. We performed an honest analysis with 1 animal equaling 1 statistical unit.

3) Reviewer 1 point 4)

Author's reply: '..... Further, we would like to point out that we disagree with the reviewer's conclusion that the immunohistochemical reactivity observed across stratum pyramidale must be artefact or "not synaptic proteins". Studies by Dodt et al. and Petralia et al. show that the synaptic subunits we examined are equally present across the stratum pyramidale as well as radiatum and oriens, particularly at the proximal dendrite. This also explains why we see a somewhat higher reactivity across the stratum pyramidale even though the somatic fraction of the examined subunits is not necessarily the most abundant one. Here, a lot of proximal dendrites arise from the somata of the stratum pyramidale and overlap. The further distal we go from the stratum pyramidale, the less dendrites overlap, leading to a less impressive reactivity pattern. While our line of reasoning would have been greatly supported by differentially regulated receptor subunits in the RNA seq analysis, we still regard our argumentation as valid: It is known that receptor composition is a highly plastic, quick and activity-driven process involving multiple levels of regulation going well beyond mere differential transcription (see e.g. Paoletti et al11). Regarding the western blot technique, it does not allow for quantification in a context specific manner, as it doesn't maintain the cellular characteristics and structure of the tissue.

In addition, the epitopes recognized by the primary antibodies in IHC may not be identically available in WB.

**** Immunohistological analysis of post-synaptic receptors, in particular NMDA receptors is not trivial. For example see Schneider Gasser 2006 Nat Prot (doi:10.1038/nprot.2006.265): .." loss of antigenicity

and/or epitope masking that are due to tissue fixation are major limitations of immunohistochemistry, notably for the detection of postsynaptic proteins. Aldehyde fixation may alter antigenic sites by cross-linking tissue proteins and antibodies into the postsynaptic density and into the synaptic cleft, a narrow space that separates the presynaptic and postsynaptic elements and is filled with extracellular matrix molecule ..". Cytoplasmic NR2A and B staining in the cell body/ prox. dendrite of PFA perfusion fixed brain largely represents a non-synaptic pool of receptors. Instead milder fixation procedures yield synaptic staining which is concentrated in the str. rad and str or (see Table 1 of Schneider Gasser 2006; and Fritch et al J Comp Neurol, 1998, 390:194–210).

It is still not clear what we see in the images of Fig. 5, and how it is connected to synaptic NMDA receptors. Does every dot reflect a psd with NMDA receptors? According to the Methods the images reflect 'a background-filtered signal indicating the approximate portion detected by the spot detection tool'. Please explain and show what type of 'background' has bin filtered by the software. Is the dotted labeling visible before 'filtering'? How is 'background' defined?

Also explain what is meant by: "reactivity levels" based on Imaris spot detection tool in the bar graphs (Figure legends of Fig. 5; y-axis of bar graphs). Do values reflect relative signal intensity, number of labeled spots, or a combination of both.

Did the authors perform Western blot? There are many NR2A/B antibodies that work on Western. If the authors want to stick to their story that NR2A/B levels are altered, Western blot would be insightful.

Images were not background-filtered before analysis in Imaris. The images presented in the manuscript figures, however, represent the post-analysis background-filtered signal to highlight the portion of signal as counted in Imaris. The "dotted labeling" as the reviewer calls it was visible before filtration and after filtration identically, see example figure attached.

We refer to "reactivity" as the number of spots normalized to volume, in this case spots/0.001mm³. 'Quality' is defined by the Imaris reference manual as "the intensity at the center of the spot in the channel the Spots were detected" (Imaris 9.2 reference manual, bitplane.com, p.432). The Imaris background subtraction tool identifies spot intensity as "(...) one of a Gaussian filtered channel from above minus the intensity of the original channel Gaussian filtered by 8/9 of spot radius (...)" (Imaris 9.2 reference manual p. 433). Intensity detection is therefore a measure relative to the local background, which proved to be most reliable in detecting the relevant portion of signal. As described in "methods", the Quality-threshold was kept unchanged throughout the analysis, allowing for an unarbitrary quantification of IHC-signal.

We completely agree with the reviewer that synaptic signal detection is not trivial. Our approach was not intended to go in as much detail as this review-discussion does now. The intention was to reliably detect an immunochemistry-based signal within the region of interest, i.e. CA1, as parallel as possible to the tissue isolation by laser dissection for RNAseq. We would like to point to our lab's previous publication² using an electron-microscopy based approach and already showing differences at a synaptic level including the NR2A/B subunit. Our present study sub-differentiates these results by analyzing NR2A and B separately, thus pointing into a future direction of research that can link transcription modulation by DNA repair enzymes to functionally relevant receptor subunit changes. We agree that a final conclusion on dynamic NR2A/B changes in this context needs further investigation, e.g. in the form of subunit-specific electron-microscopy, high-resolution confocal microscopy and/or neuronal activity dependent read-outs such as patch-RNAseq/post-stimulation IHC. This, however, is beyond the scope of this study.

We did perform Western blot analysis, however, in different, much less region-specific tissue material (manually subdissected CA1) and using different antibodies (ThermoFisher A-6473 (α -

NMDAR2A) and MA1-2014 (α -NMDAR2B)). The results were inconclusive, i.e. could neither verify nor contradict the results presented in this study. Given the limitations of this approach – especially the considerably reduced tissue specificity – we do not believe that this is a valid form of reproducing the results presented in this manuscript. A viable approach would be a new laser dissection with subsequent Western Blot or mass spectrometry; however, this is again beyond the scope of this study.

We therefore suggest presenting the NR2A/B results as preliminary with the figure as a supplement figure and a toned down conclusion in the manuscript text.

The figure will be relabeled Figure S5. Previous Figures S5, S4-1 and S4-2 will be changed to Figure S4, S3-1 and S3-2, respectively, since previous Figure S3 will be removed (for details, see answer to reviewer #3 below).

Reviewer #2 (Remarks to the Author):

The authors have addressed the comments from this reviewer to a satisfactory level, and this reviewer does not have additional concerns regarding the revised manuscript.

Reviewer #3 (Remarks to the Author):

The authors have satisfactorily addressed some but not all my comments. Particularly remaining concern is the one regarding the discrepancy between the age and sex of the mice used for behavioral studies (6 months old male mice) vs those used for electrophysiological studies (4-6 months male and female mice), and the one regarding the meaning of results showing differences in stratum oriens prevoiley and the MWZ results in face of lack of changes in LTP.

They argued that in terms of behavioral tasks (open field and elevated plus maze) a report (Shoji et al., 2016) indicated that there is no difference in performance between 4-6 months old mice. Yes, this maybe the case but this was not my question since the authors only used 6 months old male mice for the behavioral studies (all the same age and sex). My question was related to the age and sex discrepancies between the behavioral and electrophysiological studies. There are reports indicating age related changes in receptor density and calcium channels in the hippocampus (see Huang and Kandel, 2006 Learning and Memory, 13:298) and other reports on the influence of estrogen on synaptic plasticity and spatial learning (e.g., Wang et al., 2018 J Neurosci 38:7935; Yagi et al 2019 Neuropharmacol 44: 200). How can the authors compare electrophysiological data with the behavioral ones, if each one was performed in different age and sex cohorts? If the authors want to argue differently, please show that in your data set there is no sex and age differences. Alternatively, the possibility that the electrophysiological results maybe biased due to sex-dependent effect could be added to the discussion.

Added to this problem is the change in NMDA receptor composition reported by immunohistochemistry that was obtained from 3-6 months old male mice. How sure are the authors that the changes in NMDA receptor composition faithfully correlates with stratum oriens axonal activation (from females and males) and are not correlated with age and sex? How does this change in axonal activation affect the learning process, given the unaltered LTP, which has been attributed to a low NR2A/NR2B-ratio (as the authors mentioned line 268)? The authors speculates that the changes in axonal activation could be related to a decrease in density of afferents to CA1 that would be sufficient to improve spatial coding independently of increased LTP. Again, evidence that there is no sex and age difference should be provided or added in the discussion.

As stated in the previous Rebuttal letter, we agree that using mice of same age and sex in separate experiments is optimal. Thus, we have reanalyzed the electrophysiology data using results from male mice only. A statistically significant difference is still found between mutant and WT mice in the stimulation intensities needed to elicit a prevoilley of 0.5 mV, ($p = 0.0189$) (new Figure 3, vi). However, differences in stimulation intensities needed to evoke a prevoilley of 1.0 mV, are no longer significant in mutant vs WT mice; however, a tendency is still observed ($p = 0.0694$). For the 20Hz stimulation experiment (Figure S3), excluding females reduced the sample size to one male mouse per genotype and we would therefore like to remove Figure S3 from the paper.

In the paper referred to by the reviewer (Huang and Kandel, 2006) 1.5-2-month-old mice were compared to 18-month-old mice. Thus, the age-related changes in receptor density and calcium channels are perhaps not surprising as these types of changes have been shown to occur at around 8-12 months³. In the present study we have used mice that are 3-6 months of age, usually classified as mature adult⁴. This group consists of mice that are past development, but not yet affected by senescence. Several studies show that the different parameters we have analyzed in the present study do not vary or change notably in mice that are 3 to 6 months old^{3,5,6}. Thus, it is our opinion that the differences we observe between WT and *Neil1*^{-/-}/*Neil2*^{-/-} mice in behavior and in reduced axonal activation is not a result of age but of genotype.

Details regarding the age group used in the present study have been added to the Experimental model and subject section in Methods.

References

1. Aarts, E., Verhage, M., Veenvliet, J. V, Dolan, C. V & van der Sluis, S. A solution to dependency: using multilevel analysis to accommodate nested data. *Nat. Neurosci.* **17**, 491–496 (2014).
2. Regnell, C. E. *et al.* Hippocampal adult neurogenesis is maintained by Neil3-dependent repair of oxidative DNA lesions in neural progenitor cells. *Cell Rep.* **2**, 503–510 (2012).
3. Radulescu, C. I., Cerar, V., Haslehurst, P., Kopanitsa, M. & Barnes, S. J. The aging mouse brain: cognition, connectivity and calcium. *Cell Calcium* **94**, (2021).
4. Flurkey, K., Curren, J. M. & Harrison, D. E. Mouse Models in Aging Research. *Mouse Biomed. Res.* **3**, 637–672 (2007).
5. Cizeron, M. *et al.* A brainwide atlas of synapses across the mouse life span. *Science (80-)*. **369**, 270–275 (2020).
6. Lister, R. *et al.* Global Epigenomic Reconfiguration During Mammalian Brain Development. *Science (80-)*. **341**, (2013).

REVIEWERS' COMMENTS:

Reviewer #1 (Remarks to the Author):

All comments have been properly addressed

Reviewer #3 (Remarks to the Author):

This is second revision of a manuscript reporting on a study aimed to elucidate the role of two DNA glycosylases (NEIL1, NEIL2) in rodent behavior (activity, anxiety, learning and memory) and synaptic plasticity.

The previous version was criticized for not considering sex and age as biological variables. This was because the authors used only 6 months old male mice for behavioral studies, 3-6 months old males to determine by immunohistochemistry the composition of NMDA receptor subunits, and 4-6 months old males and females for electrophysiological analyses of hippocampal excitability and synaptic plasticity. In this revised manuscript the authors present a re-analysis of electrophysiological data considering only the male cohort. By doing so, statistical power seems reduced as the previously detected differences between genotypes were lost for stimulus intensity that evoked 1.0 mV prevoiley responses (Fig. 3, vi). In addition, results from previous experiments showing no significant differences in delayed response enhancements in CA3-CA1 synapses between the genotypes (Figure S3) was removed due to a significant decrease in sample size (now n=1 male mouse).

These modifications, although not affecting the overall interpretation of results, introduce a certain level of uncertainty about what would be the outcome if a larger sample size were to be used, particularly for Figure 3 where data seems to have a "tendency" (as the authors claimed) to be higher in the DKO than in WT.

REBUTTAL # 3

Hildrestrand et al.:

NEIL1 and NEIL2 DNA glycosylases modulate anxiety and learning in a cooperative manner

REVIEWERS' COMMENTS:

Reviewer #1 (Remarks to the Author):

All comments have been properly addressed

Reviewer #3 (Remarks to the Author):

This is second revision of a manuscript reporting on a study aimed to elucidate the role of two DNA glycosylases (NEIL1, NEIL2) in rodent behavior (activity, anxiety, learning and memory) and synaptic plasticity.

The previous version was criticized for not considering sex and age as biological variables. This was because the authors used only 6 months old male mice for behavioral studies, 3-6 months old males to determine by immunohistochemistry the composition of NMDA receptor subunits, and 4-6 months old males and females for electrophysiological analyses of hippocampal excitability and synaptic plasticity. In this revised manuscript the authors present a re-analysis of electrophysiological data considering only the male cohort. By doing so, statistical power seems reduced as the previously detected differences between genotypes were lost for stimulus intensity that evoked 1.0 mV prevoelley responses (Fig. 3, vi). In addition, results from previous experiments showing no significant differences in delayed response enhancements in CA3-CA1 synapses between the genotypes (Figure S3) was removed due to a significant decrease in sample size (now n=1 male mouse).

These modifications, although not affecting the overall interpretation of results, introduce a certain level of uncertainty about what would be the outcome if a larger sample size were to be used, particularly for Figure 3 where data seems to have a “tendency” (as the authors claimed) to be higher in the DKO than in WT.

In response to Reviewer #3's concerns and the editor's requests, we have performed power analyses where possible and discussed statistical issues more thoroughly in the Statistics and reproducibility section (p. 25-26) and in the Discussion (p. 13-14).

Conclusions from the electrophysiology experiments have been qualified (see Results, p. 7, and Discussion, p. 13).